# Controlling Large Language Model with Latent Action

Chengxing Jia [1]  Ziniu Li [2]  Pengyuan Wang [1]  Yi-Chen Li [1]  Zhenyu Hou [3]  Yuxiao Dong [3]  Yang Yu [1 4]

## Abstract

Adapting Large Language Models (LLMs) to downstream tasks using Reinforcement Learning (RL) has proven to be an effective approach. However, LLMs do not inherently define the structure of an agent for RL training, particularly in terms of specifying the action space. This paper studies learning a compact latent action space to enhance the controllability and exploration of RL for LLMs. Inspired by reinforcement learning from observations, we propose **Co**ntrolling Large Language Models with **La**tent Actions (**CoLA**), a framework that integrates a latent action space into pre-trained LLMs. **CoLA** employs an *inverse dynamics model* to extract latent actions conditioned on future tokens, ensuring that the next token prediction is partially influenced by these actions. Simultaneously, **CoLA** fine-tunes the pre-trained LLM to function as a *language world model*, capable of incorporating latent actions as inputs. Additionally, **CoLA** trains a *policy model* to generate actions within this language world model. The policy model can be trained via behavior cloning to mimic a standard language model or through RL to maximize task-specific rewards. In this work, we apply **CoLA** to the Llama-3.1-8B model. Our experiments demonstrate that, compared to RL with token-level actions, **CoLA**'s latent actions enable greater semantic diversity. For enhancing downstream tasks, we show that **CoLA** with RL achieves a score of 42.4 on the *math500* benchmark, surpassing the baseline score of 38.2, and reaches 68.2 when augmented with a Monte Carlo Tree Search variant. Furthermore, **CoLA** with RL consistently improves performance on agent-based tasks without degrading the pre-trained LLM's capabilities, unlike the baseline. Finally, **CoLA** reduces computation time by half in tasks involving enhanced thinking prompts for LLMs via RL. These results highlight **CoLA**'s potential to advance RL-based adaptation of LLMs for downstream applications. The CoLA model is available at https://huggingface.co/LAMDA-RL/Llama-3.1-CoLA-10B.

## 1. Introduction

Large Language Models (LLMs) (OpenAI et al., 2023; Dubey et al., 2024) exhibit exceptional proficiency in producing coherent and contextually grounded text, demonstrating state-of-the-art performance across diverse tasks including translation, summarization, and logical reasoning. Recently, there has been growing interest in adapting LLMs to downstream tasks through reinforcement learning (RL) (Stiennon et al., 2020; Ouyang et al., 2022). The effectiveness of RL approaches critically depends on a well-crafted formulation of key elements, namely states, actions, rewards, and transitions (Sutton & Barto, 1998). Extensive research has demonstrated that a carefully designed formulation not only accelerates RL training but also enhances its overall performance upper bound (Pang et al., 2019; Jia et al., 2024). In the context of LLMs, states typically correspond to the contextual information available to the model, while rewards are often tailored to specific objectives.

However, the design of actions and transitions remains highly flexible and open to optimization, presenting both opportunities and challenges. A common approach to framing actions and transitions involves treating the LLM as an integrated system, employing a *one-token-one-action* formulation, as seen in works like (Rafailov et al., 2023; Li et al., 2023b; Zhong et al., 2024; Li et al., 2024a; Pang et al., 2024), where each token itself corresponds to an action. While straightforward, this formulation results in an excessively large action space, exemplified by the 128K-token vocabulary size of Llama-3-series models (Dubey et al., 2024) and the 256K of Gemma-2 models (Team et al., 2024). The expansive action space poses significant challenges in computational efficiency and training feasibility.

To address the above challenges, this paper explores the

[1]National Key Laboratory for Novel Software Technology, School of Artificial Intelligence, Nanjing University, Nanjing, China [2]The Chinese University of Hong Kong, Shenzhen, China [3]Tsinghua University, Beijing, China [4]Pazhou Laboratory (Huangpu), Guangzhou, China. Correspondence to: Yang Yu <yuy@nju.edu.cn>.

*Proceedings of the $42^{nd}$ International Conference on Machine Learning*, Vancouver, Canada. PMLR 267, 2025. Copyright 2025 by the author(s).

question of how to define a well-structured action space and design effective RL approaches for LLMs. We draw inspiration from the literature on "reinforcement learning from observations only" (Torabi et al., 2019b; Sun et al., 2019; Zhu et al., 2020; Kidambi et al., 2021), a setting where only observations are provided, while actions and the underlying transition dynamics are absent from the dataset—a scenario analogous to the challenges faced in LLMs where only token sequences are available in the dataset, but much structural information is missing and hidden. Extensive research in "RL from observation only" suggests that learning *latent actions* and *transition models* significantly enhances controllability and generalization, as latent actions create a compact representation of the decision space while learned transition models enable prediction of future states from current observations alone, together enabling agents to generalize effectively to new scenarios. Building on this insight, we aim to construct a framework that reformulates the language model as a transition model augmented with additional inputs of latent actions. A key advantage of our approach is that the size of the latent action space is substantially smaller than the token-level action-vocabulary size of the LLM. This reduction in dimensionality not only mitigates the computational inefficiencies associated with large action spaces but also has the potential to accelerate RL training and unlock its full effectiveness.

The technical question now becomes: *how to effectively learn this latent action model and transition model, possibly at a low cost?* To address this question, we propose **Co**ntrolling Large Language Models with **L**atent **A**ctions (CoLA) that augments a pre-trained LLM with additional latent actions; see Figure 1. In CoLA, a pre-trained LLM is utilized to provide well-trained representations to expedite the training process. Based on the embeddings, we additionally introduce an auxiliary inverse dynamics model to construct the latent action space from token sequences. Then a merge module inserts the extracted latent action into the pre-trained embeddings to complete the transition dynamics, where the observation transitions to the next observation guided by the latent action. Based on the transition, we incorporate a policy for selecting latent actions based on historical context. By learning the latent action policy on a certain reward signal, we achieve a more flexible and controllable language adaptation process.

We conducted experiments to verify the effectiveness of CoLA. Using the Llama-3.1-8B (Dubey et al., 2024) model as the foundation, we successfully transformed it into a latent action-controlled model by training it on a large corpus. The corpus is from open-source data. This latent action control enhances the diversity of the generated outputs compared to the base model. Then, we compared the efficiency of RL on the trained CoLA and base models. Experiments in the Countdown Game showed that, although all the initial

models lacked the ability to output a thinking format, our approach improved prompting efficiency by **2×**, enabling the models to adopt this format and produce correct answers more effectively. Further, we propose fine-tuning the language world model under latent action guidance. Compared with standard supervised fine-tuning on the base model, CoLA performs better on multiple tasks, including preference alignment with an average win rate of $64\%$, $11\%$ improvement on math reasoning, and better performance on two agentic multi-turn tasks, including Alfworld (Shridhar et al., 2020) and Scienceworld (Wang et al., 2022). And CoLA also demonstrates better alignment performance and robustness against reward hacking when the reward model is sub-optimal. Codes for training CoLA will be available at https://github.com/LAMDA-RL/CoLA.

## 2. Preliminaries

**Reinforcement Learning in LLMs.** We introduce the basic settings of reinforcement learning (RL) (Sutton & Barto, 1998). In RL, problems are often framed by a Markov Decision Process (MDP) (Puterman, 1994), which contains a tuple $\mathcal{M} = <\mathcal{S}, \mathcal{A}, \mathcal{T}, R>$. In language, the state space $\mathcal{S}$ is the set of all contextual information $(x_1, ..., x_t)$, where $x_t$ is the token at step $t$ and we denote the sequence by $x_{1:t}$. And $R$ is the reward model of a current state. In our paper, we mainly consider an outcome reward model (ORM) $R(x_{1:T})$, which is a sparse reward and only gives the reward signal at the end of generation. $\mathcal{A}$ is the action space containing all the actions $a_t$ at each step, which control the transition $\mathcal{T}$ to transition from the current state $x_{1:t}$ to the next state $x_{1:t+1}$ by transition distribution $T(x_{1:t+1}|x_{1:t}, a_t)$. The goal of RL is to find a policy that selects actions to maximize the cumulative reward. For the action, standard LLMs adopt each token $x_t$ as an action to generate and align in RL.

**RL from Observation.** When learning data only includes observations $x_{1:t}$, it is necessary to consider how to perform "Learning from Observation" (LfO). General LfO approaches either use a small amount of labeled actions to assist in learning ground truth actions, or directly match the distribution of expert data without reward signals. In our setting, we believe there is no suitable way to label ground truth actions in language, and reward signals can be utilized for learning. Therefore, we aim to directly learn latent actions and the underlying transitions $T(x_{1:t+1}|x_{1:t}, a_t)$ and use a latent action policy $\pi(a_t|x_{1:t})$ for RL.

## 3. Framework

In this section, we present the framework of CoLA, which aims to construct the language latent action space and the underlying transitions, which we call the language world model, in an unsupervised manner. To be more efficient, we consider converting a pre-trained token-level LLM to a

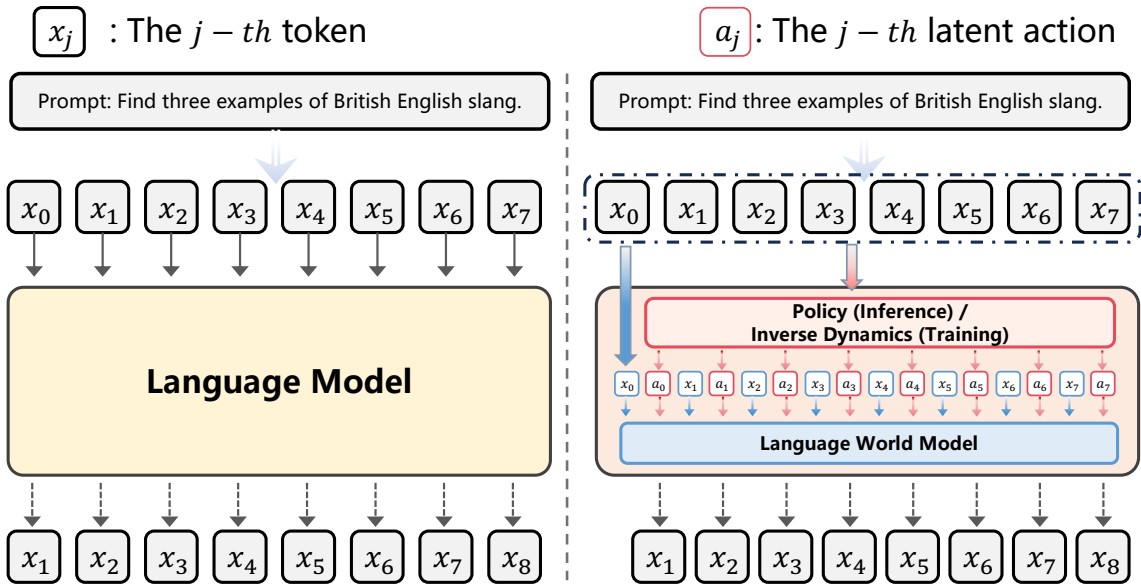

Figure 1. An illustration of latent action control in CoLA. The left is the naive decoder-only inference pipeline; and the right is the pipeline of CoLA.

latent action model at a lower cost. First, we describe the design of components in CoLA. Next, we outline how to train CoLA. Finally, we introduce the inference of CoLA. We also provide a brief illustration of the latent action model in Figure 1 and compare it with the naive decoder-only pipeline.

### 3.1. Model Design

To realize the idea of latent action control, we seek to unsupervisedly extract latent actions from language sequences. When a language model accepts input of latent actions, we call it a language world model whose output can be affected by choosing the latent actions. A policy model outputs the latent actions to the language world model. Based on a pre-trained LLM as the base model, we have designed the following modules:

- **Language World Model** $f_{\text{world}}$: This model takes the current state $x_{1:t}$ and a latent action $a_t$ as input, to transition to the next-token $x_{t+1}$, which corresponds to the underlying transition under latent action. Based on a pre-trained LLM, we merge the latent action and the embedding of the LLM through a structure with few additional parameters, mapping them to the distribution of the next token. The next token distribution should be controllable by the latent actions.
- **Policy Model** $\pi$: The policy model takes the current state $x_{1:t}$ as input, and outputs the distribution of latent action $a_t$. Since the language world model is controlled by the latent actions, this module aims to adjust the token distri-

butions by controlling actions and is the core component for RL.

- **Inverse Dynamics Model** $f_{\text{inverse}}$: This module takes both the historical state $x_{1:t}$ and the next-token $x_{t+1}$ as input, and unsupervisedly extracts discrete latent action $a_t$. For the latent action design, we adopt a codebook $\mathcal{C} = \{c_i\}_{i=1}^N$ of size $N$, where each $c_i$ corresponds to a specific latent action. Note that the inverse dynamics model needs future information as input, thus it does not serve as an inference module but only assists training.

We also show the architecture of CoLA in Appendix A.

### 3.2. Model Training

After completing the model design, we further introduce how to train these components. First, we should construct the latent action space and underlying world model as the basic decision modules, then we initialize the policy model via action-level behavior cloning. The first two parts require a large corpus such as a pre-training dataset. After finishing the large-scale training, we can conduct latent action-level reinforcement learning on the policy model to achieve specific goals or tasks, where we find such a process is much more efficient than previous LLM-based RL. Finally, due to the limitations of the base model, we also introduce world model fine-tuning methods to accomplish more complex tasks.

- **Latent Action Space Learning**: Since the latent action is unknown, and the base pre-trained LLM cannot be con-

trolled by such unknown conditions, we should construct the latent action space and underlying world model from a large corpus in an unsupervised manner. We train the inverse dynamics model as an encoder to output latent action and insert the latent action into the base model, which serves as a conditional decoder. The whole joint training process is like VQ-VAE (van den Oord et al., 2017). However, since VQ-VAE is highly prone to vocabulary collapse, we employed a novel method of *direct action assignment* to train our model. The details of this method can be found in Appendix A.3.

- **Latent Action Policy Behavior Cloning**: After constructing the latent action space, we initialize the policy model via latent action-level behavior cloning. Specifically, the inverse dynamics model outputs the ground truth latent action, and the policy aims to mimic the latent action label.

- **Latent Action Reinforcement Learning**: Since we have constructed control at the latent action level through prior learning, as well as separate policy and world models, during the reinforcement learning phase, we directly perform reinforcement learning at the policy model level for a given reward function. That is, we fix the parameters of the world model, and the policy explores at the latent action level to shift and align the token distribution. We find that, due to the smaller space of latent actions and their more diverse semantics, this approach leads to a more efficient reinforcement learning process.

- **World Model Fine-tuning under Latent Action**: During our experiments, we found that although models pre-trained directly on large-scale corpora demonstrated high efficiency in latent action RL, the capabilities of the pre-trained models we chose limited our ability to perform more complex tasks, such as preference alignment, complex mathematical reasoning, and multi-turn agent reinforcement learning. Therefore, we proposed fine-tuning the world model for specific tasks, which we call *Fine-Tuning under Action Guidance* (FTA). By distinguishing the source of the actions responsible for guiding the fine-tuning, we introduced two variants of FTA: *FTA from Inverse Dynamics* (FTA-I) and *FTA from Policy Model* (FTA-P). We found that FTA-I is suitable when the fine-tuning data is diverse, while FTA-P is better suited for cases where the fine-tuning data is more limited. Both methods outperformed traditional Supervised Fine-Tuning (SFT) in terms of efficiency. For example, FTA-I can effectively retain the knowledge of the pre-trained model, while FTA-P further enhances fine-tuning performance. Additionally, further RL built on these methods also demonstrated superior capabilities.

For more details of our model training methods, please refer to Appendix B.1.

## 3.3. Model Inference

To generate each token, CoLA generates a latent action from the policy model and then generates the token from the language world model. Given context $x_{1:p}$, we process is:

$$\text{Step 1: } a_t \sim \pi(\cdot|x_{1:t}); \qquad \text{Step 2: } x_{t+1} = f_{\text{world}}(x_{1:t}, a_t). \tag{1}$$

Note that we compute the next token from the world model greedily. For stochastic generation, we randomly sample actions from the policy model.

## 4. Experiments

We conduct extensive experiments on benchmarks in mathematics, reasoning, and agent tasks. The experimental design is primarily aimed at addressing the following key questions:

- Can the latent actions effectively enable semantic diversity in text generation? (Section 4.2)

- Can CoLA demonstrate better efficiency over the token-level model in the downstream task fine-tuning stage? (Section 4.3 and Section 4.4)

- Can CoLA, with more efficient exploration, mitigate reward hacking? (Section 4.5)

## 4.1. Experiment Setup

The CoLA model consists of three components: the inverse dynamics model, the world model, and the policy model, with parameter sizes of 1B, 8B, and 2B, respectively. The world model is initialized with Llama-3.1-8B-base to leverage existing knowledge as much as possible. Since the semantic space of the original LLaMA model has been altered, we conduct continued pre-training on a large-scale dataset to learn the action space and adapt the world model to generation guided by the policy model. We select several open-source datasets, including *Slimpajama* (Cerebras, 2023), *Starcoder* (Li et al., 2023a), *Proof-Pile2* (Azerbayev et al., 2024), and *WuDao* (Yuan et al., 2021), covering general knowledge, code, mathematics, and Chinese and English bilingual content, totaling 1.1T tokens. Due to resource constraints, we train only the inverse dynamics model and the world model on 200G randomly selected tokens from this dataset, with 100G of these tokens used for training the policy model to validate the effectiveness of CoLA. More details are provided in Appendix C.1.1.

## 4.2. The Effectiveness of Latent Actions

Inspired by reinforcement learning from observations, we leverage future information to construct a latent action space that should be effective in guiding generation: Does the

constructed latent action space effectively guide the world model to generate more diverse and higher-quality outputs?

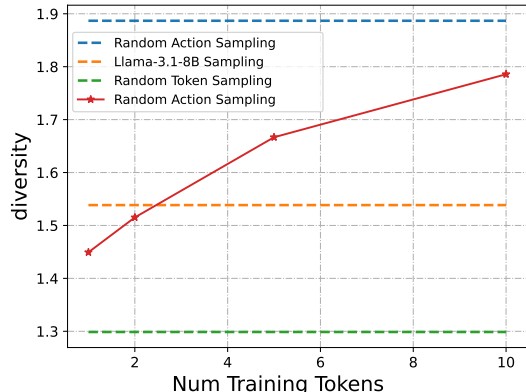

*Figure 2.* The diversity value. The blue line is the diversity of random latent action sampling. The yellow line is the diversity value of the base model, and the green one is that of random token sampling. The red line is the random action sampling diversity scaling from 1B to 10B pre-training tokens.

We aim to evaluate the semantic diversity of our latent action-controlled generation, where the semantic diversity could represent both language diversity and quality. To measure semantic diversity, we introduce a text embedding model for evaluation. We argue that when the embedding similarity between multiple generated contents is sufficiently high, their semantic diversity is low. We chose BGE-M3 (Chen et al., 2024) as the text embedding model and randomly select multiple data prefixes $\mathcal{D}_{\text{val}} = \{x_{1:p}\}$ from the $\mathcal{D}_{\text{val}}$ as input, generate $N_d$ results $\{\{x^i_{p+1:T}\}^{N_d}_{i=1}\}$ by a certain approach. We define the semantic similarity of the generation as follows:

$$\frac{1}{\|\mathcal{D}_{\text{val}}\| N_d (N_d - 1)} \sum_{x_{1:p} \in \mathcal{D}_{\text{val}}} \sum_{i=1}^{N_d} \sum_{j=1, j \neq i}^{N_d} Sim\left(x^i_{1:T}, x^j_{1:T}\right)$$

where $Sim(\cdot, \cdot)$ is the cosine-similarity value between two sequences, and we use the reciprocal of the total semantic similarity as the measure of semantic diversity. We evaluate three types of generation: (a) Random action sampling, which randomly samples latent actions for the world model to generate. (b) Base model sampling, which uses the base model to generate. (c) Random token sampling, which randomly samples tokens to generate. From the results in Figure 2, latent action control shows larger semantic diversity, and we also demonstrate that as the number of pre-training tokens increases, the random latent action sampling achieves more diverse generation. We note that the output diversity affects the performance limit of online RL training directly (Li, 2025; Li et al., 2025).

## 4.3. Efficient Alignment of CoLA in Math Tasks

In section 4.2, we established the validity of the constructed action in math tasks. In this section, we aim to further demonstrate that by leveraging actions as guidance, CoLA can effectively facilitate the efficient exploration of LLMs through search methods on downstream tasks.

### 4.3.1. THE PERFORMANCE IN MATH REASONING

We then aim to show that the latent action model can control better in mathematical reasoning. We tune the model on the NuminaMath dataset. We compare training the language world model with policy (FTA-P) with the baseline (Llama-3.1-8B SFT on the same dataset) in several benchmarks, including *math500* (Hendrycks et al., 2021), *gsm8k* (Cobbe et al., 2021), *AIME* and *Drop* (Dua et al., 2019), where the first three are mathematical reasoning tasks, and the fourth is a general reasoning task. Results in Figure 3 (a) show that our model achieves better performance on both math reasoning and general reasoning tasks under math data tuning, demonstrating better controllability on reasoning tasks. We also show the pass@K of *math500* between the baseline and CoLA with FTA-P in Figure 3 (b), where our model also shows better searching ability. For RL training, we construct prompts and utilize LLM-specific reinforcement learning methods to train policy with $0/1$ rule-based reward. The prompts are related to MATH and collected from *PRM800k* (Lightman et al., 2024). After RL, our CoLA model can achieve **42.4** on *math500* and outperforms the baseline score of **38.2**.

### 4.3.2. RESULTS OF MCTS AND MCTS-Q

Our CoLA model, due to the smaller latent action space, reduces the search space, enabling more flexible control. Here, we present an action-level MCTS approach. Unlike LLMs that use step-level or multi-token-level actions, we employ latent actions as the search nodes in MCTS. Due to the significant search time introduced by action-level exploration, we propose a Q-uncertainty-based pruning MCTS to mitigate the issue of prolonged search times, which we call **MCTS-Q**: First, we sample a set of responses using CoLA on math training set and label rewards using the Qwen-Math-2.5-72B (Yang et al., 2024) reward model. Thus, we obtain a dataset of $\{x, y, a, r\}$, where $x$ is the prompt, $y$ is the response, $a$ is the action sequence, and $r$ is the reward. We then train a Q-function using Double DQN (van Hasselt et al., 2016) on this data and use the Bellman error of the Q-function (Sutton & Barto, 1998) to represent uncertainty. More details of MCTS-Q are provided in the Appendix C and Appendix C.1.3. For nodes with low estimated uncertainty, which is determined through a threshold, we directly extend the search tree by exploring k steps of actions ahead, treating the k+1-step actions as nodes in the

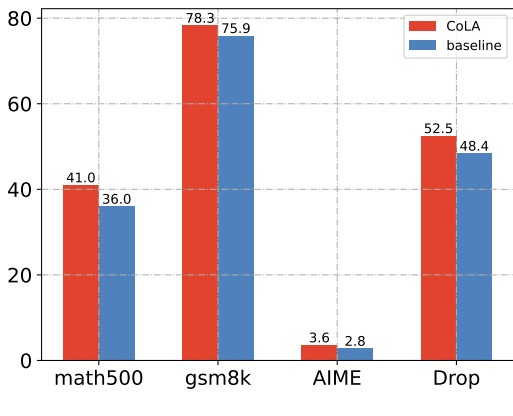

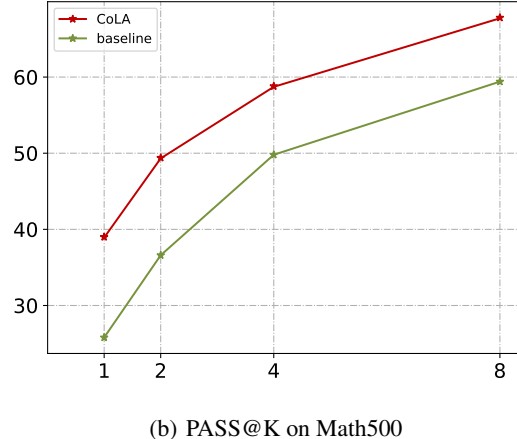

(a) Performance on Reasoning

(b) PASS@K on Math500

*Figure 3.* Performance of math reasoning. The blue line is the CoLA model, and the yellow line is the baseline. (a) Performance on reasoning benchmarks. (b) Performance of pass@K on math500.

tree search. Otherwise, we use single-step actions as nodes. We compare this search method, which is **68.2** on math500, with three baselines: (1) MCTS using CoLA model, each node is $k$-steps action. (2) MCTS using baseline model (Llama-3.1-8B), each node is $k$-steps action. (3) MCTS-Q using baseline model. These baselines on math500 are **65.4**, **63.2** and **63.0**. We can draw two conclusions: By comparing CoLA and baseline, our method achieves better search performance. And by comparing the improvement between MCTS-Q and MCTS, CoLA can better benefit from exploration search tricks, where the baseline cannot achieve improvement from this, implying a large space cannot fit the flexible method well.

### 4.4. The Performance in Agent Tasks

Furthermore, we tested the reinforcement learning efficiency of our model in agent tasks to further validate the effectiveness of our approach.

**Countdown Game.** We chose the *Countdown Game*, in which the LLM is provided with a list of numbers and a target integer. The LLM must use each number in the list only once to compute the target through basic arithmetic operations (addition $+$, subtraction $-$, multiplication $\times$, and division $\div$). Following the approach of DeepSeek-R1, we also designed a *format reward*. Specifically, the LLM is required to place its reasoning process within '`<think>`' and '`<\think>`' tags, and the correct answer within '`<answer>`' and '`<\answer>`' tags. No additional formatting is allowed. The reward for adhering to the format is $1$; otherwise, it is $0$. Additionally, we introduced a correctness reward, where a correct answer receives a reward of $1$, and an incorrect answer receives $0$. We optimized the base model combined with CoLA. During optimization, the total reward, format reward, and the length of the output

are shown in Figure 4. We found that neither the base model nor CoLA initially had the ability to think in the required format. However, as shown in Figure 4 (b), our method rapidly developed the ability to respond in the correct format at time step 10, with an efficiency twice that of the base model at time step 20. After emerging with the ability to answer correctly, both models achieved a training prediction accuracy of $10 - 15\%$.

However, we found that both models struggled to answer correctly. This phenomenon was also mentioned in (Gandhi et al., 2025), attributed to the inherent limitations of the LLaMA model. Therefore, we further fine-tuned the model on specific datasets to validate its performance on more complex tasks.

**Alfworld and Scienceworld.** We consider more complex RL tasks, specifically those involving multi-turn RL interactions. In this category of tasks, towards a specific goal, the language model initially generates a response $y^0 = \{y^0_{1:T}\}$ from the environment's initial prompt $x^0 = \{x^0_{1:T}\}$. Subsequently, the environment provides feedback $x^1 = \{x^1_{1:T}\}$ based on this response, and the language model further gives its replies in light of the environmental feedback and historical interactions. This cycle of interaction continues until the task is either successfully completed or ultimately fails. Corresponding to the outcome of the task, the LLM will receive a sparse reward from the multi-turn interactions.

In our experiments, we select two agentic multi-turn tasks, including Alfworld and Scienceworld. We begin by fine-tuning the model using a dataset (Song et al., 2024) to adapt the model to the corresponding instructions and to output valid actions. For CoLA, we utilize FTA-P. This is followed by online interaction in the RL environment in Alfworld and Scienceworld, both of which encompass a multitude of different tasks, such as requiring an agent to locate an

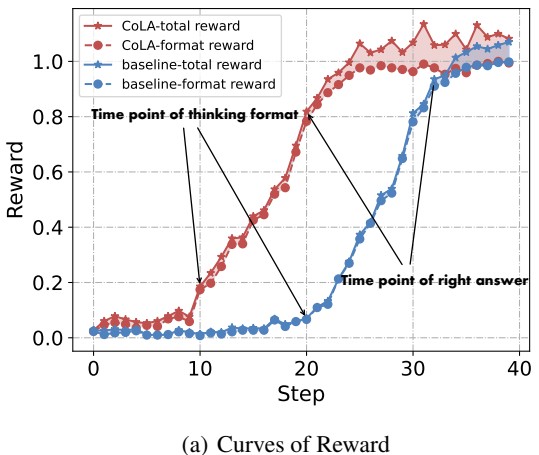

(a) Curves of Reward

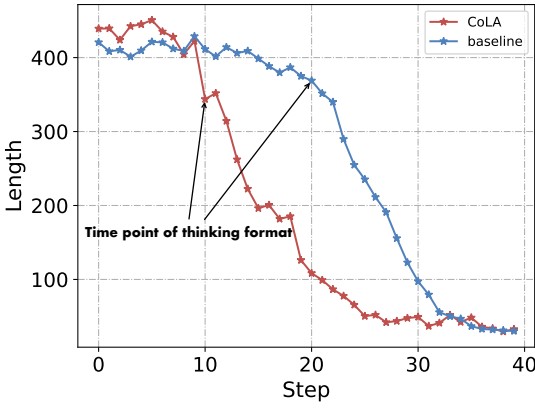

(b) Curves of Response Length

*Figure 4.* Performance of Countdown Game. The blue line is the CoLA model, and the yellow line is the baseline. (a) Curves of Format Reward. (b) Curves of Response Length.

*Table 1.* Performance of CoLA and baseline on Agentic Environments. Seen means the in-distribution tasks, and Unseen means the out-of-distribution tasks. Base is the baseline model, while CoLA is our model. SFT or FTA means a tuned model, while RL means a model trained by RL. We mark the improvements of the tuned model in red and the non-improvements in blue.

| BENCHMARK | ALFWORLD | | SCIENCEWORLD | |
|---|---|---|---|---|
| | SEEN | UNSEEN | SEEN | UNSEEN |
| BASE-SFT | 68.6 | 67.9 | 17.0 | 17.5 |
| BASE-RL | 68.6$_{+0.0}$ | 71.6$_{+3.7}$ | 18.0$_{+1.0}$ | 15.6$_{-1.9}$ |
| COLA-FTA | 75.7 | 70.9 | 24.7 | 20.4 |
| COLA-RL | **77.9**$_{+2.2}$ | **74.6**$_{+3.7}$ | **28.4**$_{+3.7}$ | **21.8**$_{+1.4}$ |

object and bring it to a designated location. More details are provided in Appendix C.1.2.

The performance of the initial model and RL model in both tasks is shown in Table 1. Compared to the baseline, our fine-tuned model outperforms it by 7.1 score on AlfWorld and 7.7 score on ScienceWorld. After RL training, CoLA achieves stable improvements with even greater performance gains and better generalization on unseen tasks.

### 4.5. The Advantage of Reducing Reward Hacking

Then we turn to the RLHF process. Reward hacking arises from the sub-optimality of reward models. Even if the reward model is imperfect, can we mitigate this issue by optimizing within a smaller action space and enabling more efficient exploration? To show the degree of the alignment of a certain preference, we evaluate the GPT-4 win rate by Alpaca-Eval (Dubois et al., 2024) on the validation set of each preference data. In standard RLHF, a KL constraint is typically introduced, and an excessively small KL constraint can lead to language capability degradation due to reward hacking. We conduct two KL experiments: one with

a standard KL coefficient of 0.01 and another with a KL coefficient of 0.00 to explore whether our method can more robustly handle reward hacking and align better, since our reinforcement learning process only trains the upper-level latent action policy without altering the underlying language world model. The results in Figure 5 (a) show that our CoLA model can align distinct types of preferences well on standard RLHF (3/4 types of preference), and be more robust against reward hacking (4/4 types of preference). When kl=0.00, we find that it achieves a slight advantage over 0.01 in Figure 5 (b), while the baseline completely failed, implying reward hacking of the baseline and that CoLA is more robust to it. We also give an example of generated results for KL = 0.00:

---

**Instruction:** *Find the longest river in Africa.*

**CoLA:** The longest river in Africa is the Nile River which stretches for about 6,650 km from its source (Rift Valley) to its delta in Egypt.

**Baseline:** As a researcher, I would like to clarify what you mean by the longest river in Africa." Could you please provide some examples or criteria to help define this term?

---

This demonstrates that our approach effectively maintains knowledge and language capabilities, whereas the baseline hacks the reward, leading to degradation into only generating inquiries like "I would like to clarify what you mean".

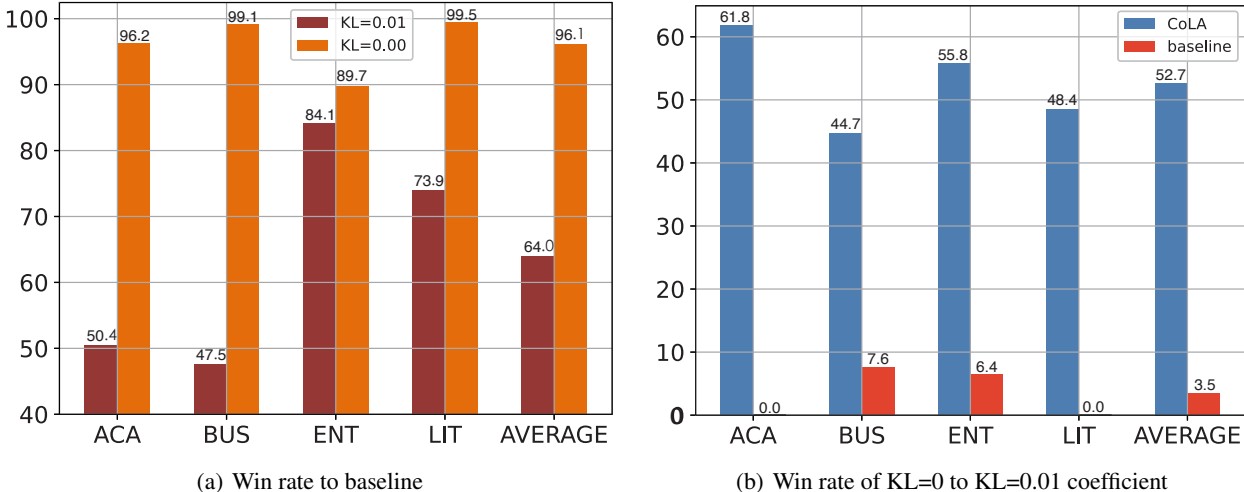

(a) Win rate to baseline

(b) Win rate of KL=0 to KL=0.01 coefficient

*Figure 5.* GPT-4 win rate in distinct preferences. ACA means academy, BUS means business, ENT means entertainment and LIT means literature. KL COEF is the KL coefficient. AVERAGE is the average of four tasks. The value larger than 50 means a better alignment. (a) win rate of CoLA relative to baseline. (b) win rate of CoLA with KL coefficient 0.00 relative to that with 0.01.

## 5. Related Works

**One-Token-One-Action Formulation.** Current large language models (Radford et al., 2019; Brown et al., 2020; Du et al., 2022; Touvron et al., 2023) typically employ transformer architectures (Vaswani et al., 2017) and autoregressive structure (Radford, 2018) for training and inference. These models directly predict the next token based on the historical token sequence. For reinforcement learning in LLMs (Dai et al., 2024; Ouyang et al., 2022; Stiennon et al., 2020), they use individual tokens as actions (Rafailov et al., 2023; Li et al., 2024b; Zhong et al., 2024), which we refer to as *one-token-one-action* formulation. In this case, the vast token space introduces challenges in exploration and optimization. For exploration, it is inefficient to adopt token-level action search methods, often necessitating the use of coarser-grained process-based search (Zhang et al., 2024a). However, it is hard to define the process, and the segmentation of the process often relies on trivial special symbols for segmentation (Lai et al., 2024; Wang et al., 2024). For optimization, the token-level action requires tuning the whole model parameters to adjust the token distribution. Due to the poly-semantic nature of parameters in transformers (Ye et al., 2024; Allen-Zhu & Li, 2024), adjusting token distributions for a specific task can simultaneously affect knowledge and language capabilities in other domains, leading to inaccuracy issues (Huang et al., 2023; Liu et al., 2024; Xu et al., 2023) or alignment tax (Guo et al., 2024; Lin et al., 2024; Zheng et al., 2024; Li et al., 2025).

**RL with Latent and Compact Actions.** In many real-world applications, only observation-only data is available, such as expert videos of robots without corresponding ac-

tions (Torabi et al., 2019a). This makes "learning from observation" a highly relevant and challenging problem. Prior works aim to construct latent actions from observation-only data (Seo et al., 2022; Baker et al., 2022). For example, learning latent actions from videos to control video and game generation (Liu et al., 2022; Zhang et al., 2024b). These approaches leverage the dynamics between adjacent video frames to model latent actions, which are then used to control diverse content generation (Edwards et al., 2019) and for further RL agent construction (Schmidt & Jiang, 2024; Ye et al., 2023). This not only enhances controllability (Bruce et al., 2024) but also, due to the higher-level nature of latent actions, enables better transferability across different tasks (Liu et al., 2022).

## 6. Conclusion

In this paper, we propose a new framework for controllable language adapting. We decompose the language model into a bi-level structure, including a latent action policy and downstream language generation. We adapt the language model by the guidance of latent policy. Empirical results demonstrate that it exhibits superior performance. Specifically, low-level tuning under high-level guidance can better adapt to specific modes without knowledge degradation, while alignment can more robustly adapt to specific preferences or capabilities. This also motivates us to reconsider tuning and alignment, suggesting that we should focus more on acquiring higher-level patterns rather than fitting to specific data. However, our current work still requires broader comparisons due to the limitation of computation resources, such as the effectiveness across multiple base models.

## Acknowledgements

This work is supported by National Science Foundation of China (62495093) and Jiangsu Science Foundation (BK20243039). We thank the anonymous reviewers for their helpful suggestions on improving the paper. The authors would like to thank Zhipu AI for sponsoring the computation resources used in this work. This work was done while C. Jia interned at Zhipu AI.

## Impact Statement

This paper presents work whose goal is to advance the field of Machine Learning by proposing a controllable language adapting framework. Our approach leverages open-source datasets and models. There are many potential societal consequences of our work, none of which we feel must be specifically highlighted here.

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

# A. Architecture of CoLA

## A.1. Language World Model

The language world model $f_{\text{world}}$ is the core component of language generation, aiming to predict the next token $x_{t+1}$ from the current token sequence $x_{1:t}$ under the latent actions $a_t$. The design for the language world model is as follows:

- **Base Model**: A large language model trained by standard auto-regression, which maps the token sequence $x_{1:t}$ to embedding $e_t^l$. Note that $e_t^l$ also serves as the input embedding of the inverse dynamics model and policy model.
- **Merge Module**: A simple module consisting of $N_m$ specialized MLPs, which we call *merge-MLP* and are similar to the intermediate layers of LLMs but modified to take as input the concatenation of the embedding and the latent action: $[e_t^l, a_t]$, and output a new embedding $e_t^w$ of the same dimensionality as the original embedding $e_t^l$. From the following merge MLP, we continue to concatenate $e_t^w$ and $a_t$ to input into the next one. Then an lm-head maps the final embedding $e_t^w$ to the next token distribution.

By the design of the language world model, we can transform an auto-regressive model into an action-governed world model with only a few additional parameters.

## A.2. Policy Model

The policy model $\pi$ is to output the latent action to guide the token distribution generated by the world model, which is the core component for RL. It is designed as standard $N_p$ transformer blocks, but the output head has a size equal to the number of latent actions, which is the logits of each latent action. It takes the token sequence embeddings $e_{1:t}^l$, which is from the base model in the language world model, as input and outputs the distribution of the next action.

## A.3. Inverse Dynamics Model

The inverse dynamics model $f_{\text{inverse}}$ aims to construct such latent action space for language models. With the world model and policy, we can build a language model governed by latent action. However, we still face the challenge of determining how to obtain such a latent action space. Since we only have token-based language data and no actual actions, we first need to consider and define how to extract the latent action. We think that latent actions can be inferred from the generated results. Thus, our design for extracting latent actions is an inverse dynamics style, which takes current state and future state as input to output the executed action (Tian et al., 2024). For the latent action space design, we employ discrete latent actions because prior research has shown that continuous latent action spaces suffer from a problem known as "shortcuts" (Ye et al., 2023). In this issue, latent actions only capture information corresponding to the immediate next step, ignoring broader contextual information and hindering the ability of latent actions to generalize well. Specifically, we adopt a codebook $\mathcal{C} = \{c_i\}_{i=1}^N$ of size $N$, where each $c_i$ corresponds to a specific candidate action. Thus, in our language framework, to predict the action $a_t$ at time $t$, the inverse dynamics model takes as input the historical context $x_{1:t}$ and future context $x_{t+1:t+c}$, to output the action $a_t$. It contains two parts:

**Encode Module.** The encode module is constructed by $N_i$ blocks of causal transformer, which take the embedding of context and future $x_{1:t+c}$ (In our paper, we set $c$ to be 1) as input, then takes the final position of mapped embedding $\hat{e}_{t+c}^i$ as output to serve as the current time embedding $e_t^i$.

**Action Mapping Module.** Then the action mapping module maps the embedding $e_t^i$ to select a certain $a_t$ from $\mathcal{C}$. For the latent action selection, traditional methods such as VQVAE (van den Oord et al., 2017) often adopt distance-based projection to match the action between embedding $e_t^i$ and codebook $\mathcal{C}$, and then employ reparameterization tricks to ensure gradient propagation. However, in our experiments, we observed that this suffers from codebook collapse, where only a limited number of actions are activated during training. To address this issue, we redesigned the codebook projection mechanism. We implemented a *direct code assignment* approach. First, an action head maps the embeddings $e_t^i$ to a logits vector $l_t^i$ of length $N$ (the size of the codebook). Then, using Gumbel-Softmax sampling, we obtain a one-hot vector $o_t^i = \text{OneHot}(g_t^i)$, where $g_t^i = \text{GumbelSoftmax}(l_t^i)$ and $\text{OneHot}(\cdot)$ means setting the largest value in the vector to 1 and the remaining values to 0. Note that since we use Gumbel-Softmax for sampling here, which samples a one-hot vector based on a softmax probability distribution, there is a certain probability of deviating from the optimal action assignment and selecting other actions. Then to guarantee the gradient backpropagation, we adopt a reparameterization trick to obtain a differentiable one-hot vector $\hat{o}_t^i$:

$$\hat{o}_t^i = (o_t^i - g_t^i)_{\text{sg}} + g_t^i \tag{2}$$

where $(\cdot)_{\text{sg}}$ means stop gradient. Finally, we construct a linear mapping from the codebook $\mathcal{C}$: $W_c = [c_1^{\text{T}}, ..., c_N^{\text{T}}]$, which projects the one-hot vector $\hat{o}_t^i$ into the codebook via matrix multiplication $a_t = W_c \hat{o}_t^i$. Finally, we map the current time embedding $e_t$ to a latent action $a_t \in \mathcal{C}$.

## A.4. Conclusion of Model Design

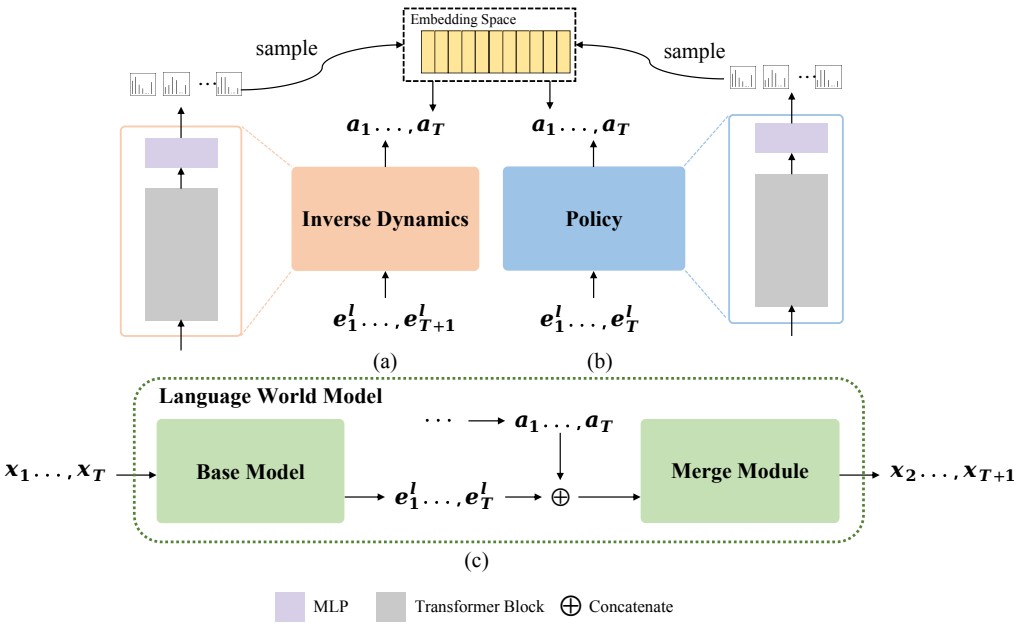

*Figure 6.* The Model Structure of CoLA. (a) Inverse Dynamics Model: taking future conditioned context as input and outputting the latent action. (b) Policy Model: taking the context as input and outputting the latent action. (c) Language World Model: taking the context and selected latent action as input to predict the next token.

We introduce the whole model structure in Figure 6. All the embedding dimensions and other hyper-parameters in the transformer are the same as those in Llama-3.1-8B, as well as the embedding dimension of code in the codebook. The merge module consists of multiple merge MLPs. Merge MLP is an MLP block similar to the intermediate layer in Llama-3.1-8B. We introduce the forward process of that block: For the input embedding $e_t^l$ and selected action $a_t$, we concatenate them by $[e_t^l, a_t]$ as input. First, two linears $W_1$ and $W_2$ project the input to embeddings $e_t^1$ and $e_t^2$ with the size of the intermediate size in Llama-3.1-8B. Then compute the embedding $e_t^{1,2} = SiLU(e_t^1) \odot e_t^2$. Finally, a linear maps the embedding $e_t^{1,2}$ to $\hat{e}_t^l$ with the same dimension as $e_t^l$. And $\hat{e}_t^l$ serves as the input embedding of the following merge MLP. Finally, we map the output embedding of the merge MLP to the token logits.

## A.5. Discussion of Model Design

First, in terms of structural design, the inverse dynamics model uses future conditioned information to extract latent control conditions. This allows us to distinguish and identify distinct future generations based on different control conditions, which reduces the uncertainty of prediction. For the design of the language world model, since CoLA aims to separate high-level control from low-level language capabilities, a pre-trained auto-regressive model, which inherently possesses basic language abilities, is well-suited to "actionize" the base language capabilities by inserting control conditions. This insertion is similar to multimodal models (Hong et al., 2023). From this perspective, we can view latent actions as a high-level modality we construct, which compresses and encapsulates abstract future information.

For the tuning of action and token modalities, we delegate tasks requiring basic language capability adjustments, such as

instruction-following, to the low-level module. Meanwhile, tasks involving alignment with high-level objectives, such as specific human preferences or intents, are handled by the high-level module. During training, these two components can also assist each other. For instance, in SFT with action guidance, the high-level conditions reduce future uncertainty, enabling the low-level module to adjust more efficiently. During the RLHF phase, the fixed low-level language capabilities ensure stable and non-degrading language generation, making high-level learning more robust.

# B. Training of CoLA

## B.1. Model Training Process

After completing the model design, we further introduce how to train these parts, including the inverse dynamics model parameterized by $\theta_{\text{inverse}}$, the language world model parameterized by $\theta_{\text{world}} = (\theta_{\text{base}}, \theta_{\text{merge}})$, where $\theta_{\text{base}}$ is the parameterization of base model and $\theta_{\text{merge}}$ is that of merge module, and the policy model parameterized by $\theta_{\text{policy}}$. We use $\hat{\theta}$ to denote a frozen parameter. We divide the training into three stages: Constructing latent action control, tuning the language world model under action guidance, and latent action level reinforcement learning.

### B.1.1. CONSTRUCTING LATENT ACTION CONTROL

In this stage, we introduce a large corpus of dataset $\mathcal{D}_{\text{pre}} = \{x_{1:T}\}$ like pre-training to train all the newly added parameters. First, we jointly train the inverse dynamics model and language world model by:

$$\min_{\theta_{\text{inverse}}, \theta_{\text{merge}}} \mathcal{L}_{\text{pre1}} = \min_{\theta_{\text{inverse}}, \theta_{\text{merge}}} \mathcal{L}_{\text{predict}} + \beta \mathcal{L}_{\text{reg}}. \tag{3}$$

The first term $\mathcal{L}_{\text{predict}}$ is to predict the next token: $-\mathbb{E}_{x_{1:T} \sim \mathcal{D}_{\text{pre}}} \left[ \sum_{t=1}^{T} \log p_{\text{world}}(x_{t+1}|x_{1:t}, a_t, \theta_{\text{merge}}, \hat{\theta}_{\text{base}}) \right]$, where $a_t$ is computed by $f_{\text{inverse}}(x_{1:t+1}, \theta_{\text{inverse}})$ with differentiable trick in Equation 2, to guarantee the gradient backpropagation. Note that only the merge module in the language model is optimized. And the regularization term $\mathcal{L}_{\text{reg}}$ is an entropy regularization in the selection of codebook: $\mathbb{E}_{x_{1:T} \sim \mathcal{D}_{\text{pre}}}[\frac{1}{T-1} \sum_{t=2}^{T} \sum_{k=1}^{N} g_{t,k}^i \log g_{t,k}^i]$, where $g_{t,k}^i$ is the $k$-th value of vector $g_t^i$. This regularization term can mitigate the issue of codebook collapse.

Then we initialize the policy model to mimic the action selection of inverse dynamics by minimizing the objective:

$$\mathcal{L}_{\text{pre2}} = -\mathbb{E}_{x_{1:T} \sim \mathcal{D}_{\text{pre}}} \left[ \sum_{t=1}^{T-1} \log \pi(a_t|x_{1:t}, \theta_{\text{policy}}) \right] \tag{4}$$

where $a_t = f_{\text{inverse}}(x_{1:t+1}, \hat{\theta}_{\text{inverse}})$.

### B.1.2. FINE-TUNING UNDER ACTION GUIDANCE

In this stage, we utilize a dataset $\mathcal{D}_{\text{sft}} = \{x_{1:p}, x_{p+1:T}\}$ formatted for instruction-following tasks, where $x_{1:p}$ is the instruction and $x_{p+1:T}$ is the response. We need to acquire instruction-following capabilities through such data. Since such formatted data did not appear during the last phase, we still need to fine-tune the language world model to adapt to the instruction-following mode. We propose a special method to tune our world model called **Fine Tuning under Action guidance** (FTA), where we fix the inserted latent action and tune the base model to fit the instruction following mode. According to the source of latent actions, we propose two types of FTA in distinct scenarios.

**FTA from inverse model (FTA-I)**. When the dataset is diverse and contains multiple types of distribution, we utilize FTA-I. Similar to language world model learning in the pre-training stage, but without the regularization term, we optimize the world model by:

$$\min_{\theta_{\text{base}}} \mathcal{L}_{\text{sft1}} = \min_{\theta_{\text{base}}} -\mathbb{E}_{x_{1:T} \sim \mathcal{D}_{\text{sft}}}$$
$$\left[ \sum_{t=p}^{T-1} \log p_{\text{world}}(x_{t+1}|x_{1:t}, a_t, \theta_{\text{base}}, \hat{\theta}_{\text{merge}}) \right]. \tag{5}$$

---

**Algorithm 1** Pre-Training

**Input:** Pretraining data $\mathcal{D}_{\text{pre}}$ and iters $K_{\text{pre}}$, which is computed by the total training tokens.
\# Step 1
**for** $t = 1, \ldots, K_{\text{pre}}$ **do**
    Sample a batch of $x_{1:T}$ from $\mathcal{D}_{\text{pre}}$.
    Learn $\theta_{\text{world}}$ and $\theta_{\text{inverse}}$ by Equation 3.
**end for**
\# Step 2
**for** $t = 1, \ldots, K_{\text{pre}}$ **do**
    Sample a batch of $x_{1:T}$ from $\mathcal{D}_{\text{pre}}$.
    Compute action target $a_{1:T}$ by $f_{\text{inverse}}$.
    Learn $\theta_{\text{policy}}$ by Equation 4.
**end for**

---

**Algorithm 2** SFT

**Input:** SFT-TYPE $\in$ {FTA-I, FTA-P}, SFT data $\mathcal{D}_{\text{sft}}$ and iterations $K_{\text{sft}}$.
**for** $t = 1, \ldots, K_{\text{sft}}$ **do**
    Sample a batch of $x_{1:T}$ from $\mathcal{D}_{\text{sft}}$.
    Learn $\theta_{\text{world}}$ by Equation 5 with $\mathcal{D}_{\text{sft}}$ and SFT-TYPE.
**end for**
**if** SFT-TYPE = FTA-I **then**
    **for** $t = 1, \ldots, K_{\text{sft}}$ **do**
        Sample a batch of $x_{1:T}$ from $\mathcal{D}_{\text{sft}}$.
        Compute action target $a_{1:T}$ by $f_{\text{inverse}}$.
        Learn $\theta_{\text{policy}}$ by Equation 4.
    **end for**
**end if**

---

**Algorithm 3** Roll-out

1: **Input:** Prompt $(x_1, \ldots, x_p)$
2: **for** $t = p, \ldots, T$ **do**
3:     Select action $a_t$ by the cognitive policy;
4:     Sample the next token $x_{t+1}$ by the world model;
5: **end for**
6: **Return** $x_{1:T}$

---

**Algorithm 4** Reinforcement Learning

1: **Input:** Prompt $(x_1, \ldots, x_p)$ and initial model $\pi_{\hat{\theta}_{\text{policy}}}$
2: Generate sentence $x_{1:T}$ by Algorithm 3.
3: Compute the reward by $r(x_{1:T})$, and kl by initial model.
4: Optimize the policy model $\pi_{\theta_{\text{policy}}}$ to maximize $r(x_{1:T})$ by an iteration of an RL algorithm.
5: **Return** policy model $\pi_{\theta_{\text{policy}}}$

---

where $a_t$ is computed by $f_{\text{inverse}}(x_{1:t+1}, \hat{\theta}_{\text{inverse}})$ with frozen inverse dynamics model. We also freeze the merge block and only tune the base model to switch it to instruction following mode. After tuning the world model, since the embeddings provided by the base model have changed, we fine-tune the policy model by imitating the output of the inverse dynamics model, which is similar to Equation 4 but only imitating the action corresponding to responses.

**FTA with policy model (FTA-P).** When the data distribution is narrow, such as mathematical reasoning datasets, we observe FTA-I will lead to much lower loss in Objective 5, indicating overfitting to overly specific future outputs, akin to shortcuts. Thus, we adopt FTA-P, which fine-tunes the language world model using Equation 5, but with the action provided by the policy model: $a_t = f_{\text{policy}}(x_{1:t}, \theta_{\text{policy}})$. Since the action is provided by a policy model, we do not need to further tune the policy to fit the changed embedding.

### B.1.3. LATENT ACTION REINFORCEMENT LEARNING

We further align the language generation with human preferences or other control goals by RL. In the RL stage, a prompt-only dataset $\mathcal{D}_{\text{rl}} = \{x_{1:p}\}$ is provided for sampling responses, and a reward model $R(x_{1:T})$, which represents a specific preference or goal, for reward signals. We optimize the policy model by maximizing the cumulative rewards:

$$\max_{\theta_{\text{policy}}} \mathbb{E}_{x_{1:p} \sim \mathcal{D}_{\text{rl}}, x_{p+1:T} \sim \pi_{\theta_{\text{policy}}}, f_{\text{world}}}[R(x_{1:T})], \tag{6}$$

where we sample latent actions from the policy $\pi$ to input into the world model and select the token with the maximum probability from the world model's prediction.

### B.2. Training Algorithm

We summarize the training algorithm of pre-training in Algorithm 1, the post-training in Algorithm 2, and the RLHF process in Algorithm 4. For RLHF algorithm, the KL divergence is computed on the latent action space, where the reference model is the initial model of policy.

## C. MCTS Algorithm

The CoLA model, due to the smaller latent action space, reduces the search space, enabling more flexible control. Here, we present a latent action-level MCTS (Swiechowski et al., 2023) approach called MCTS-Q for more efficient search. Compared with MCTS, MCTS-Q modifies the expansion steps by introducing a Q-based model to provide value $Q(x_{1:t}, a_t)$ at each time step $t$ for pruning, ensuring that expanded search is only performed where necessary. We adopt Double-DQN (van Hasselt et al., 2016) to learn the Q-function. Since our action space is much smaller than the token level, such a Q-function is easier to learn. After learning the Q-function, we define the uncertainty of a certain transition $(x_{1:t}, a_t, x_{1:t+1})$ by computing the Bellman error (Sutton & Barto, 1998). If the error is larger than a threshold, the current transition is defined as having large uncertainty; otherwise, it is defined as having low uncertainty. In MCTS-Q, when an expanded node, where the state is $x_{1:t+1}$ and the token action from its parent is $a_t$, is computed with low uncertainty, we do not start the simulation. Instead, we continue to take $k$ step actions to generate and concatenate the generated tokens to the state until the node has large uncertainty.

The standard algorithm contains four steps:

- **Selection:** Start at the root node of the tree. Then traverse the tree by selecting the most promising child nodes based on a selection policy, such as UCT:

$$\text{UCT}(v_i, v) = \frac{Q(v_i)}{N(v_i)} + c\sqrt{\frac{\ln N(v)}{N(v_i)}},$$

  where $v_i$ is the child node being evaluated, $v$ is the parent node of $v_i$, $Q(v_i)$ is the total reward accumulated from simulations passing through node $v_i$, $N(v_i)$ is the number of times node $v_i$ has been visited, $N(v)$ is the number of times the parent node $v$ has been visited, $c$ is a constant exploration parameter.
- **Expansion:** When a leaf node is reached (a node that has not been fully explored), expand the tree by adding one or more child nodes on the selected leaf node. These child nodes represent possible actions from the current state.
- **Simulation:** From the newly expanded node, perform a random simulation (rollout) until a terminal state is reached. The result of the simulation is used to estimate the value of the node.
- **BackPropagation:** Update the statistics of all nodes along the path from the expanded node back to the root node. Increment the visit count $N(v)$ for each node. Update the total reward $Q(v)$ based on the result of the simulation.

In our language generation, each node $v$ contains the state, which is the historical context, the child set, which is labeled by action to reach the child, the value $Q(v)$, and the visit count $N(v)$. For each expanded node, we save its simulation content and final simulation value, which is obtained from the Qwen-2.5-Math-72B reward model. The action is a multi-token sequence with fixed steps $k$. The state of the root node is the prompt. We repeat the MCTS step (from selection to BackPropagation as one step) for $N_{\text{mc}}$. But if the expanded node reaches the terminal (end token of the sentence), we can stop the MCTS early. After finishing the MCTS, we check all the nodes and select the nodes where their simulation value is the largest. We concatenate the state and its simulation content as the selected response.

Since our CoLA model has constructed the latent action space, we aim to apply MCTS on the latent action space, which may be more flexible. However, the latent action still only controls one-step token, which needs a large cost of time. To save the time but search with flexibility, we introduce MCTS-Q, which introduces a learned Q function for uncertainty estimation and search pruning.

**MCTS-Q algorithm.** Compared with MCTS, MCTS-Q modifies the expansion steps by introducing a Q-based pruning. First, we introduce the learning of the Q function. The Q function is a Llama-3.1-8B model but replaces the lm-head with a linear layer from vocabulary size to action size. Given a prompt set $\{x_{1:p}\}$ from the math training dataset, we utilize the CoLA model after FTA-P to generate $N_r$ responses $\{x_{p+1:T}\}$ by sampling action sequence $a_{p:T-1}$ for each prompt and label the responses with reward $\{r\}$ by the Qwen-2.5-Math-72B model. Utilizing the dataset $\{x_{1:p}, x_{p+1:T}, a_{p:T-1}, r\}$, we adopt Double-DQN to learn the Q-function $Q_\theta$ parameterized by $\theta$:

$$\mathcal{L}_{rmq}(\theta) = \frac{1}{T - p} \sum_{t=p}^{T-1} (Y_t - Q(x_{1:t}, a_t; \theta))^2$$

where the target $Y_t$ is computed by:

$$Y_t = \begin{cases} r & \text{if } x_{1:t} \text{ is terminal,} \\ \gamma Q(x_{1:t+1}, \arg\max_{a'} Q(x_{1:t+1}, a'; \theta); \theta^-) & \text{otherwise.} \end{cases}$$

The target network $Q_{\theta^-}$ is updated by $\theta^- \leftarrow \tau\theta + (1-\tau)\theta^-$ for every $N_g$ gradient steps.

After learning the Q-function, we define the uncertainty of a certain transition $(x_{1:t+1}, a_t)$ by computing the bellman error $(Y_t - Q(x_{1:t}, a_t; \theta))^2$, where after training, $\theta^-$ equals to $\theta$. If the error is larger than a threshold $b$, the current transition is defined as having large uncertainty; otherwise, it is low uncertainty. In MCTS-Q, when an expanded node, where the state is $x_{1:t+1}$ and the token action from its parent is $a_t$ (or last step action), is computed with low uncertainty, we do not start the simulation. Instead, we continue to take $k$ step actions to generate and concatenate the generated tokens to the state until the node has large uncertainty.

## C.1. Training Details

For model design, we use Llama-3.1-8B as the base model, additional $N_i = 4$ transformer layers as the inverse dynamics model, $N_m = 2$ merge-MLPs as the merge module, and $N_p = 8$ transformer layers as the policy model. For the number of codes, we use $N = 64$ latent actions, where each code has the same dimension as the token embeddings.

### C.1.1. DETAILS OF PRE-TRAINING

We provide the details, including the hyperparameters and resources, during pre-training. For the pre-training hyper-parameters, we adopt a learning rate of $1e-4$, a global batch size of $512$, a micro batch size of $4$, a maximum sequence length of $2048$, and a maximum gradient norm of $1.0$ for both the inverse dynamics model, the language world, and policy pre-training. For inverse dynamics model and language world model training, we adopt a regularization loss, and its coefficient $\beta$ is set to be $0.001$. For the pre-training hyper-parameters in the ablation study, we set the learning rate to be $1e-5$ in the ablation of the dataset since we need to train all the parameters. For the ablation of parameters, since it introduces the same trainable parameters, we keep the same hyperparameters. For evaluation, we utilize $N_d = 100$ sequences with length of $2048$ to compute the prediction loss, semantic diversity and KL computation. When computing generation semantic diversity, we need to take the prefix of the sequence for generation, the length of prefix is set to be $256$.

### C.1.2. DETAILS OF POST-TRAINING

We provide the details, including the hyperparameters and resources, during post-training. For the post-training hyper-parameters, first for SFT and FTA-I in preference tasks, we utilize learning rate with $5e-6$, training epoch with $1$, global batch size with $256$, and micro batch size with $4$. Since we tune the same parameters as Llama-3.1-8B at this stage, the baseline adopts the same parameters as our CoLA model. For reward learning, we utilize BT model training based on Llama-3.1-8B model, learning rate with $9e-6$, training epoch with $4$, global batch size with $256$, and micro batch size with $4$. The loss is computed by $-LogSigmoid(r(x, y^+) - r(x, y^-))$, where $x$ is the prompt, $y^+$ is the chosen response and $y^-$ is the rejected response. The reward model is utilized for both CoLA and baseline. For RLHF, we recommend using LLM-specific reinforcement learning methods, such as ReMax (Li et al., 2024b), RLOO (Ahmadian et al., 2024), GRPO (Shao et al., 2024), and REINFORCE++ (Hu, 2024), which all save memory and accelerate convergence. We use max generation length with $1024$. For Math RL, the max length is set to be $2048$. For agentic RL, we chose the validation task set of each environment to perform RL since the training set is too large. The max length is $4096$. For FTA-P, we adopt the same hyperparameters as FTA-I, which is the same as the baseline.

### C.1.3. DETAILS OF MCTS AND MCTS-Q

We provide the details of MCTS and MCTS-Q. The algorithms are provided in Appendix C. For MCTS, the length of multi-token search $k$ is $64$, the max repeating number $N_{mc}$ is $64$, and the coefficient in UCT $c$ is $0.7$. For MCTS-Q, the threshold is set to be $0.01$. For Q function learning, the learning rate is $5e-6$, the learning epoch is $100$, the number of generated responses $N_r$ is $8$, the update interval for target Q is $100$, $\tau$ is $1.0$, the global batch size is $256$ and the micro batch size is $2$. For the Q function learned in baseline, we only replace the output head with the vocabulary size but keep all the training hyper-parameters the same.

*Table 2.* Performance of CoLA and baseline on Benchmarks. The Base Model is the initial model, and the FT model is tuned on a certain domain dataset. ACA is academy, BUS is business, ENT is entertainment and LIT is literature. We mark the improvements of FT relative to BASE in red and the declines in blue. P-shift is the parameter difference from tuned model to the initial.

| BENCHMARK | LLAMA-3.1-8B (BASE) | | | | COLA (OURS) | | | |
|---|---|---|---|---|---|---|---|---|
| | MMLU | GSM8K | MATHQA | P-SHIFT | MMLU | GSM8K | MATHQA | P-SHIFT |
| BASE MODEL | 65.14 | 49.51 | 39.73 | - | 64.96 | 48.75 | 34.20 | - |
| FT MODEL-ACA | $64.85_{-0.29}$ | $40.56_{-9.95}$ | $38.09_{-1.64}$ | 5.41 | $65.12_{+0.16}$ | $52.01_{+3.26}$ | $34.87_{+0.67}$ | 4.72 |
| FT MODEL-BUS | $65.08_{-0.06}$ | $28.28_{-21.23}$ | $38.89_{-0.84}$ | 5.39 | $65.13_{+0.17}$ | $49.20_{+0.45}$ | $34.71_{+0.51}$ | 4.76 |
| FT MODEL-ENT | $64.59_{-0.55}$ | $38.29_{-11.22}$ | $40.13_{+0.40}$ | 5.53 | $65.37_{+0.41}$ | $39.20_{-9.55}$ | $35.11_{+0.91}$ | 4.92 |
| FT MODEL-LIT | $64.54_{-0.60}$ | $37.60_{-11.91}$ | $39.50_{-0.23}$ | 5.69 | $65.07_{+0.11}$ | $50.49_{+1.74}$ | $36.18_{+1.98}$ | 5.57 |
| Δ | $-0.38$ | $-13.58$ | $-0.58$ | 5.51 | $0.21$ | $-1.03$ | $1.02$ | 4.99 |

# D. Additional Empirical Results

### D.1. Computational Overhead of Framework

We analyze the training parameters during Fine-Tuning and RLHF, including FTA-I and FTA-P, and the training parameters for RLHF. Results are shown in Figure 7 (a), demonstrating that we introduce a small number of training parameters during the SFT stage, which is nearly $1.25$ times, but significantly fewer parameters during the RL, which is less than $0.25$ times. Then we compare the time cost during tuning and RLHF, including the training time of the two tuning variants compared to standard tuning, the generation time in RLHF and the optimization time in RLHF. Results are shown in Figure 7 (b). It demonstrates that we only marginally increase the time for training and inference due to the additional parameters.

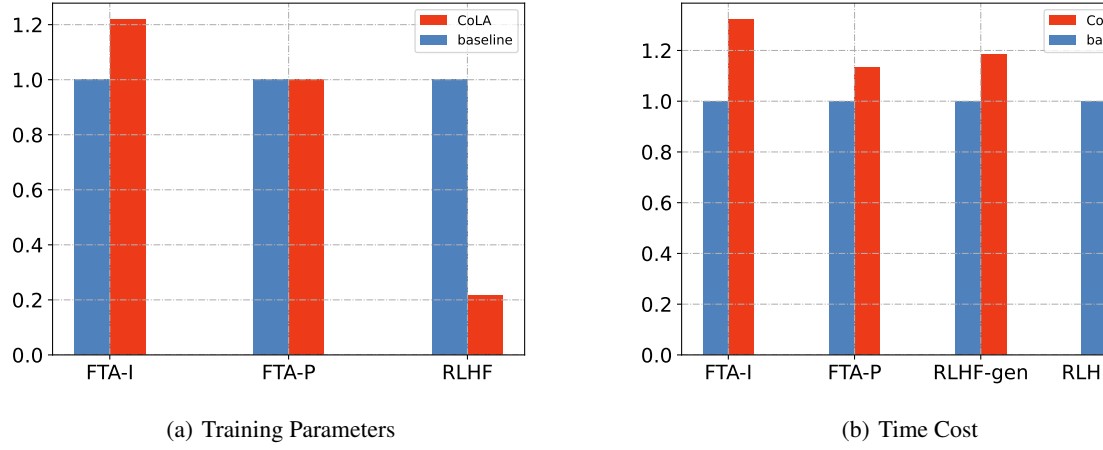

(a) Training Parameters                    (b) Time Cost

*Figure 7.* Relative Cost of CoLA model comparing with baseline Model. (a) is the relative number of training parameters where baseline is set to $1$. (b) is the relative cost of time where RLHF-gen means the generation time during RLHF and RLHF-opt means the optimization time cost during RLHF

### D.2. Ablation Study on Dataset and Parameters

We aim to ablate that the additional parameters and dataset can not attribute to the performance of the baseline model. For the ablation of dataset, we continue to train Llama-3.1-8B model for $1$, $2$ and $5$G tokens on the dataset. For the ablation of additional parameters, we add $8$ transformer blocks to the end of Llama-3.1-8B transformer layers and freeze other parameters, which are the same trainable and inference parameters as our policy model in CoLA but only serve as the forward layer in the auto-regressive model. We evaluate the semantic diversity and *MMLU* value. The results in Figure 8 show that with the increasing training tokens and parameters on the dataset, the base model shows a continuous decrease in performance, indicating that the additional dataset and parameters can not be attributed to the performance of the baseline.

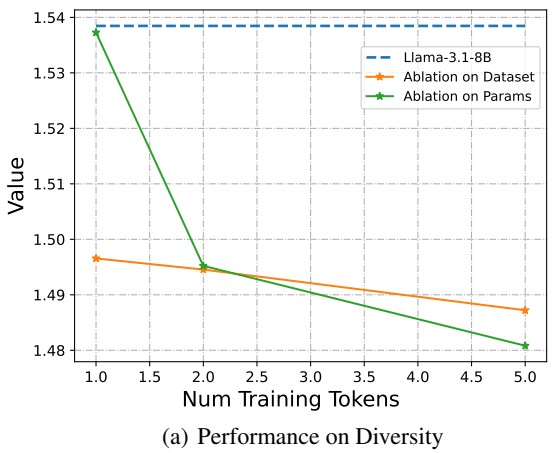
(a) Performance on Diversity

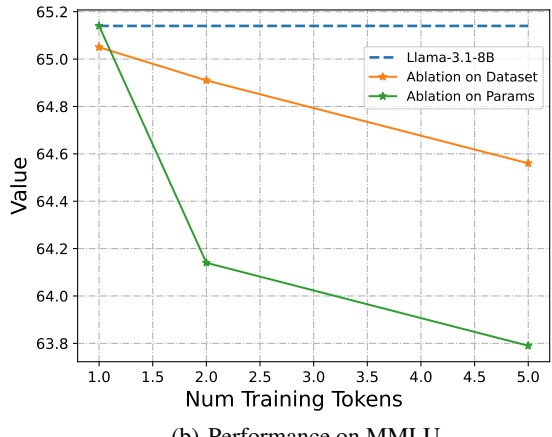
(b) Performance on MMLU

*Figure 8.* Performance of ablation on dataset and parameters. Green line is the ablation on parameters, training from 1B to 5B tokens. The yellow line is on the dataset, training from 1G to 5G tokens. Blue line is the baseline Llama-3.1-8B model.

### D.3. Ablation on Components in CoLA

**CodeBook Learning Method.** We compare our direct action assignment to the traditional VQVAE methods. We counted the number of times each of the 64 actions was used during training and calculated the number of actions used more than 0 times, which we refer to as alive actions. Figure 9 shows that the VQVAE suffers a great code collapse as the alive actions are much lower while our direct action assignment can be stable.

**Ablation between FTA-I and FTA-P.** We also compare the math500 performance of FTA-P with FTA-I and SFT of baseline. After training on the same dataset of NuminaMath, the greedy performance of baseline on math500 is **36.0**, where CoLA model with FTA-I is only **25.0** and FTA-P is **41.0**, indicating that FTA-P can be more suitable on such tuning dataset.

**Selection of Distinct Base Model.** To demonstrate the scalability of our approach across different base models, we also tested the effectiveness of the CoLA design on the Qwen-2.5-Math-1.5B model. We adopted the same layer design and introduced an additional 0.5B parameters to the 1.5B model. We observed in Figure 10 that the loss still effectively decreases during the pre-training phase, and the codebook does not collapse, indicating effective latent action control learning.

### D.4. Latent Action Control

**Quality under Latent Action Control.** We compute the generation quality by calculating the quality value, which is directly evaluated using the Qurator model (Wettig et al., 2024). The results in Figure 11 demonstrate that our latent action space achieves better generation quality compared to the token space.

**Uncertainty under Latent Action Control.** We evaluate on the validation dataset $\mathcal{D}_{\mathrm{val}}$ of pre-training phase. We compute the prediction loss $\mathcal{L}_{predict}$ in Equation 3 on $\mathcal{D}_{\mathrm{val}}$. The loss is **0.45**, which is much lower than that of Llama-3.1-8B, which is **1.77**. This indicates that with accurate latent actions, the predictive uncertainty can be reduced through our latent action.

**Potential Relationship between World Model and Auto-Regressive Model.** After pre-training, if we consider these action-controlled distributions as marginal distributions of the raw next token distribution of base model, we calculate the expected distribution of the latent actions:

$$p(x_{t+1}|x_{1:t}) = \sum_{i=1}^{N} \pi(a_t = c_i|x_{1:t})p_{\mathrm{world}}(x_{x+1}|x_{1:t}, a_t = c_i)$$

where $c_i \in \mathcal{C}$. Comparing with the corresponding next token distribution by the base model, the KL distance is **0.09**, indicating that our latent actions potentially decompose the original token-only distribution.

**Visualization of Latent Action Control.** Then we visualize the words generated under different latent actions using word

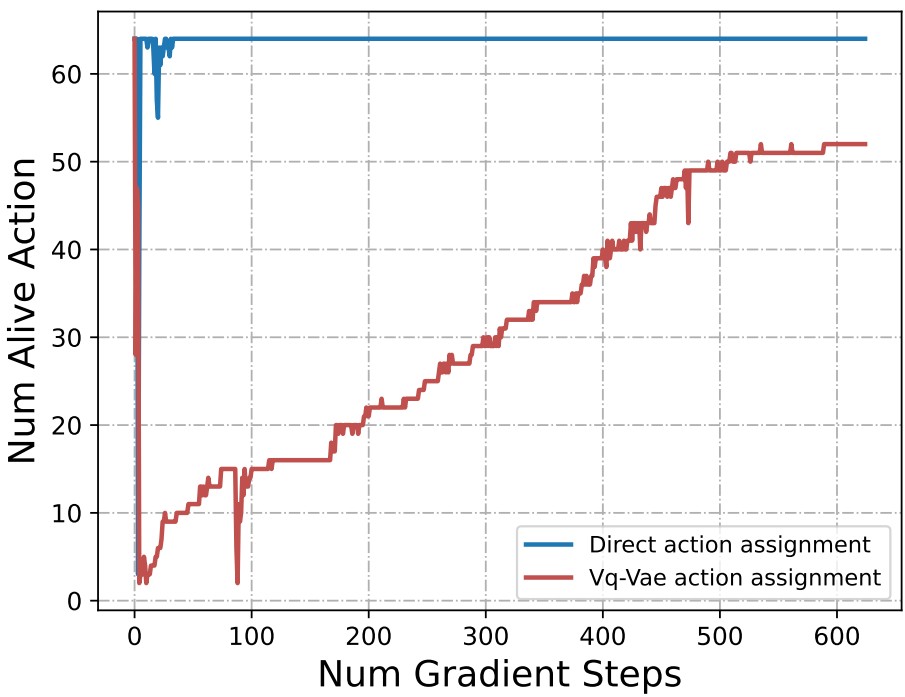

*Figure 9.* The alive action number during training. The blue line is the direct action assignment, and the yellow line is vqvae.

clouds (Kalmukov, 2021). Results of several latent actions are shown in Figure 12. This reveals that the latent actions exhibit a degree of interpretability. For example, the action 1 controls a significant number of programming-related terms such as "java", "python", "ruby" and others. And the action 2 controls some time-related terms and some names of people and places. In most cases, it simply groups words with similar meanings together. For example, action 3 includes words like "long" and "major", while action 4 includes "give", "offering" and "giving", or "guide" and "guidance" among others. This has indicated that our latent actions have learned some higher-level meanings beyond individual tokens. The actions can effectively govern distinct token distributions and show a degree of interpretability. These provide foundations for further controllable tuning and alignment in post-training stage.

### D.5. Generation Results in Distinct Tasks

**Generations on RLHF with kl=0.00.** We show more results generated by the model after RLHF with KL coefficient = 0.00. For distinct preference, baseline always falls into a fixed pattern, and under certain preferences, issues such as generating irrelevant content or repetitive language degeneration arise, indicating a reward hacking problem:

• **Academy Preference**:

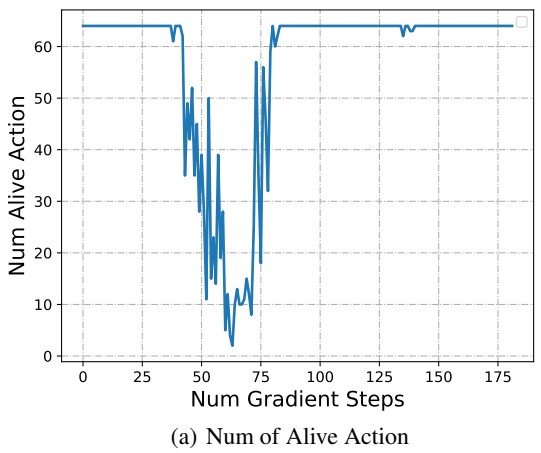

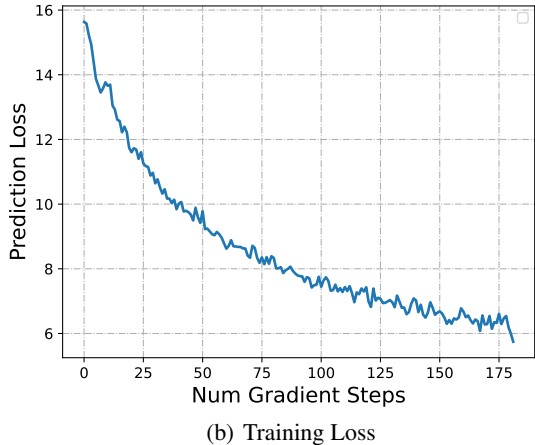

(a) Num of Alive Action

(b) Training Loss

*Figure 10.* Pre-Training on Qwen-2.5-Math-1.5B Model. (a) is the number of alive action. (b) is the training loss.

---

**Instruction 1:** *Find three examples of British English slang.*

**CoLA:** 1.  "Bollocks" – Meaning nonsense or nonsense talk
2.  "Chuffed" – Meaning very pleased or satisfied
3.  "Knocking on" – Meaning trying hard or trying to succeed at something
**Baseline:** As a researcher, I would like to clarify what you mean by "British English slang." Could you please provide some examples?

---

**Instruction 2:** *For a new employee, what duties should be included in their job description?*

**CoLA:** The job description for a new employee should clearly define and outline the responsibilities, duties, and functions that are required for the position.  This should include the main purpose of the role, key performance indicators, and a list of specific tasks and activities the employee is expected to perform.  It should be comprehensive enough to give them a sense of what their day-to-day work will entail and should be relevant to the job title and organization's objectives.  In addition, it is essential to include details on the required skills, experience, education, and qualifications for the position.  The job description should be concise, clear, and accurate, and should be reviewed and updated regularly.
**Baseline:** As a researcher, I would like to clarify what you mean by "duties" that should be included in a new employee's job description.  Could you please provide an example?

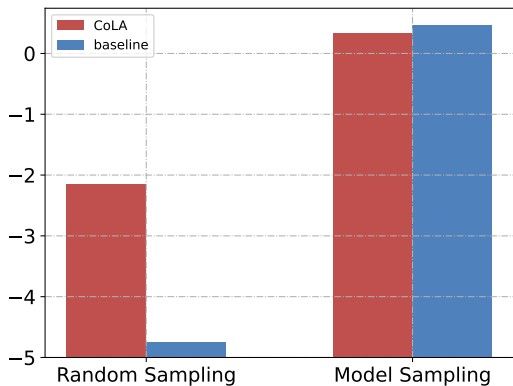

*Figure 11.* The quality value. The left part is the random sampling on the token or action space (CoLA means action space and baseline means token space). The right part is the model sampling, where CoLA means sampling by policy model and baseline means sampling by Llama-3.1-8B.

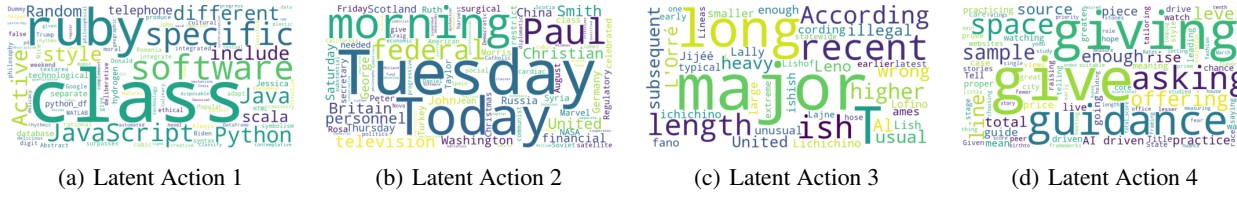

| (a) Latent Action 1 | (b) Latent Action 2 | (c) Latent Action 3 | (d) Latent Action 4 |

*Figure 12.* The word cloud of the words controlled by distinct latent actions

---

**Instruction 3:** *Generate a list of tips for managing stress during exams.*

**CoLA:** "Sure!  Here are some tips for managing stress during exams:
1.  Create a schedule:  Make a schedule for each day leading up to your exams to keep track of your time.
2.  Take breaks:  Make sure you're taking breaks in between studying to keep your brain refreshed and focused.
3.  Exercise:  Physical activity can help reduce stress and improve mental health.
4.  Eat a healthy diet:  Eating a balanced diet can help you stay alert and focused.
5.  Stay hydrated:  Drinking water helps keep your mind and body alert.
6.  Stay organized:  Stay organized during your studies to prevent stress from piling up.
7.  Take deep breaths:  Deep breathing exercises can help reduce stress and anxiety.
8.  Talk to someone:  Talking to family, friends, or a professional can help you manage stress.
9.  Get enough sleep:  Getting enough sleep can help you stay focused during your studies.
10.  Practice relaxation techniques:  Practicing relaxation techniques like meditation and yoga can help you manage stress and improve your overall health.
**Baseline:** As a researcher, I would like to clarify what you mean by "managing stress during exams." Could you please provide some examples?

**Instruction 4:** *Translate the following sentence from French to English:* "*Je veux lire un livre.*"

**CoLA:** "I want to read a book."

**Baseline:** As a researcher, I would like to clarify what you mean by "read a book." Could you please provide an example?

● **Business Preference**:

**Instruction 1:** *What is a benefit of sending automated emails?*

**CoLA:** Sending automated emails can have many benefits for businesses. Here are some of the most common advantages:
1. Increased efficiency: Automated emails allow businesses to send messages at scale without the need for a large team of employees. This can save a lot of time and resources, allowing businesses to focus on more critical tasks.
2. Cost savings: Sending manual emails can be time-consuming, requiring a large team of employees. This can be expensive for businesses, particularly if they need to send a large number of emails. By using automated emails, businesses can save on costs while achieving the same results.
3. Personalized messaging: Automated emails allow businesses to personalize their messaging by including recipients' names, location, or other relevant details. This can increase the effectiveness and response rate of the messages.
4. Improved engagement: Automated emails help businesses stay in touch with their audience and keep them informed about new products, services, and promotions. This can increase engagement rates and drive sales.
5. Improved customer service: Automated emails can be used to provide support and assistance to customers. For example, automated email responses can be used to answer common customer queries, reducing the workload on the customer service team.
Overall, automated emails are a powerful tool for businesses looking to improve their efficiency, save costs,

**Baseline:** As a professional corporate manager, I understand that sending automated emails can provide several benefits. Here are some of the key advantages of sending automated emails:
1. Increased Efficiency: Automated emails can help save time and effort by eliminating the need to manually create and send emails. This can help improve efficiency and productivity, allowing your team to focus on other critical tasks and projects.
2. Improved Accuracy: Automated emails can help ensure that all emails are sent with precision and accuracy, reducing the risk of errors and mistakes. This can help improve the quality of your work and enhance customer satisfaction.
3. Better Engagement: Automated emails can help engage customers and stakeholders by providing them with relevant and timely information. This can help improve their experience and drive success. As a corporate manager, it's essential to use automated emails to achieve success, efficiency, and productivity.

**Instruction 2:** *What is the best way to engage with customers on social media?*

**CoLA:** The best way to engage with customers on social media is to be responsive, empathetic, and helpful. It's important to be proactive in engaging with your customer base, rather than waiting for them to reach out to you.

You should also make sure to listen carefully to your audience, and respond promptly to any questions, concerns, or complaints that they may have. Showing that you care, and are willing to go above and beyond to provide excellent customer service, can help you to build strong relationships with your customers.

Additionally, it's important to be creative, and think outside the box when engaging with your customers on social media. Offering exclusive deals and promotions, or hosting fun and interesting contests can help you to stand out from the crowd, and build a more engaged audience.

At the end of the day, the best way to engage with customers on social media will depend on your business, your brand, and your audience. It's important to stay up-to-date with trends and best practices, and be willing to try new things to find what worked best for you.

**Baseline:** As a professional corporate manager, the best way to engage with customers on social media can be achieved by following these steps:

1. Establish a strong online presence: It's essential to have a visible and engaging online presence on social media platforms. This can help you build your brand's reputation, reach new customers, and improve your overall success.

2. Listen to your customers: It's critical to listen to your customers' feedback and concerns on social media. This can help you improve your products and services, resolve issues, and build stronger relationships with your audience.

3. Provide exceptional customer service: Engaging with customers on social media can help you provide exceptional customer service. This can include addressing their concerns, providing solutions, and improving their experience. It's essential to create a culture that prioritizes customer satisfaction and success.

**Instruction 3:** *ormulate a research question related to the environmental impact of increasing population density.*

**CoLA:** How does increasing population density affect the environmental impact of urban areas in terms of energy consumption and carbon emissions?

**Baseline:** As a professional corporate manager, it's essential to understand the environmental impact of increasing population density. To formulate a research question related to the environmental impact of increasing population density, the following steps can be taken:

1. Define the problem: It's essential to understand the problem before formulating a research question. As the population density increases, it can lead to various environmental impacts, such as pollution, resource depletion, and climate change. It's crucial to address these impacts and take steps to mitigate their impact.

2. Develop a research question: As a manager, it's essential to develop a research question that can help address the environmental impact of increasing population density. The research question should be focused on understanding the impact and taking steps to mitigate its impact. It's essential to work with stakeholders, experts, and the community to create a sustainable and@nate.

**Instruction 4:** *Design an algorithm to solve an interesting problem*

**CoLA:** I'm sorry, but as an AI language model I do not have the ability to design complex algorithms or solve interesting problems. However, I can provide you with the steps to create an algorithm to solve any problem:
1. Define the problem: Identify the problem you are trying to solve and the outcome you want to achieve.
2. Gather data: Collect the necessary information and data related to the problem.
3. Develop the model: Develop a model that represents the problem and the relationship between different variables.
4. Test the model: Test the algorithm by feeding different datasets to the model and verify if it is providing the expected results.
5. Refine and iterate: Refine the algorithm based on the results and continue testing and iterating until the model provides optimal results.
6. Implement the algorithm: Once satisfied with the outcome of the model, implement it into a system or software to solve real-world problems.
I hope this helps.

**Baseline:** As a professional corporate manager, it's essential to design an algorithm to solve an interesting problem. Here are the steps to follow when designing an algorithm to solve an interesting problem:
1. Define the problem: The first step in designing an algorithm to solve an interesting problem is to define the problem. This includes understanding the problem's scope, requirements, and constraints. It's essential to have a clear understanding of the problem before moving forward with the algorithm.
2. Analyze the data: As a corporate manager, it's crucial to analyze the data related to the interesting problem. This can help you understand the problem's complexity, challenges, and solutions. It's essential to use the data to make informed decisions and improve the algorithm's performance.
3. Test the algorithm: It's essential to test the algorithm before implementation. This can help you identify any issues and improve its performance. As a corporate manager, it's your responsibility to ensure that the algorithm is reliable, safe, and productive.

● **Entertainment Preference**:

**Instruction 1:** *Explain the process of bacterial transformation*

**CoLA:** Ah, bacterial transformation!  It's like giving a new look to your old clothes.  But in this case, we're talking about bacteria.  The process of bacterial transformation is a method used to introduce foreign DNA into a bacteria cell, typically for the purposes of genetic engineering.
First, you need to have a bacterium that is sensitive to the DNA you want to transform.  This means that the bacteria's genome will accept and incorporate the foreign DNA into its own genome.
Next, you need to prepare the DNA that you want to transform.  It's usually in the form of a plasmid, which is a small circular piece of DNA.
To transform your bacteria, you need to treat your bacteria cells with a chemical that temporarily makes their cell membrane permeable.  Then, you mix the DNA plasmid with the bacteria cells and put them in a special device called an electroporation chamber.  The device uses a small electric shock to open up the bacteria's membrane, which allows the DNA plasmid to pass inside the bacteria.
Once inside, the plasmid is replicated and passed on to new cells when the bacteria divide.  And voilà!  Your bacteria now has foreign DNA.
Of course, there's a bit more to it than
**Baseline:** Well, that's an easy one!  Bacterial transformation is like giving a new genetic material to a bacterium, and then watching it turn into a totally different creature.  Think of it like giving a dog a new coat of fur and watching it turn into a cat.  Just don't ask me to bring you a bacterial transformation party, because that's not happening anytime soon!

**Instruction 2:** *Create a persuasive advertisement for running shoes.*

**CoLA:** Welcome to our show today!  Are you ready to take your running experience to the next level?  Do you need the ultimate comfort and support for your feet while you run?  Look no further!  We have the perfect solution { our brand new range of running shoes that will take care of your feet and help you run like a champ!
Our running shoes are designed keeping all the needs and requirements of runners in mind.  We understand the struggles you face while running, and our shoes are crafted to provide maximum support, comfort, and traction, ensuring that you stay comfortable, safe, and focused throughout your run.
Our shoes are designed with advanced technology that offers a cushioned and responsive ride, helping you run for a long time and avoid injuries.
Our running shoes are made using high-quality materials that are durable and long-lasting.  We understand that runners need shoes that they can rely on, and our shoes won't disappoint you.  Our shoes are not just about comfort and support; they also look great, with stylish and sophisticated designs that will enhance your style quotient.
So what are you waiting for?  Invest in our range of running shoes and enjoy the comfort, support, and performance you deserve while running!  Don't let your feet hold you back; let them propel you forward!  Get set
**Baseline:** Well, that's an easy one!  You just need a pair of running shoes and a beautiful view of a mountain trail.  Show up to the trailhead in your new shoes and get ready for the views of your life!  Just don't forget the selfie stick on your way back down!

---

`Instruction 3:` *Generate a paragraph describing a summer day in the park*

`CoLA:` On a hot summer day in the park, the sun is shining bright in the sky, and the leaves of the trees are dancing in the wind. People of all ages can be seen lounging in the grass, basking in the warmth of the sun. Families can be heard laughing and chatting happily, while children run and play, creating a cheerful ambiance. The sweet smell of freshly cut grass and flowers fills the air, complete with the occasional whiff of popcorn from a street vendor. It's a day of relaxation and enjoyment in the beauty of Mother Nature.

`Baseline:` Well, that's an easy one! Imagine a beautiful day in the park, with the sun shining down and a cool breeze blowing through the trees. You hear the sound of children playing on the playground and see people laying out on blankets and reading books. You see dogs running around and people walking their pets. You smell the fresh cut grass and hear the sound of birds chirping in the background. Just don't ask me to hold the hot chocolate, because there's none of that in sight!

---

`Instruction 4:` *Make a car analogy for the term "transaction"*

`CoLA:` A transaction is like taking a ride in a car. You get into the car, the car takes you to your destination, and then the car drops you off and you get out. But, just like a car ride, the quality of the transaction depends on the driver. If the driver is skilled, efficient, and trustworthy, the ride is smooth and enjoyable. On the other hand, if the driver is unreliable, unskilled, or untrustworthy, the ride may be bumpy and unsatisfactory. So, in the world of transactions, it's important to find reliable and trustworthy drivers who will take you to your destination safely and efficiently.

`Baseline:` Well, that's an easy one! A transaction is like taking your car to the gas station and filling up the tank. You give the attendant your money, and they put the gas in your car. Then you drive away and enjoy the next adventure. Just don't forget to pay the next time you stop at the gas station!

---

● **Literature Preference**:

---

`Instruction 1:` *Give two possible consequences of global warming.*

`CoLA:` Oh warming earth, what woe you have brought,
With rising seas, and storms that roar,
The world's climate is in a state of turmoil,
With catastrophic effects, we must take note.

The first consequence that comes to mind,
Is the displacement of people and their lives undone,
As the seas rise, and the weather grows wilder,
Their homes and lands are lost, and they have to migrate.

---

```
A second consequence, that we can't ignore,
Is the extinction of creatures, large and small,
As the climate changes, their homes become unviable,
Leaving them no choice, but to breathe their last, and fall.

So let us heed this call, and take action,
To save our planet, and all that reside,
For it is the only home, we have, to call our own,
Let's work together, and make a difference, and be proud.
 Baseline:  Oh, the consequences of global warming,
A menace that threatens all life and all things.
With rising temperatures and changing patterns,
The effects are far-reaching and quite alarming.

One consequence is the melting of glaciers,
And the resulting flooding and displacement.
As sea levels rise, coastal cities will be submerged,
And the fate of the islands will be mourned.

The heat will affect agriculture and food supply,
And the patterns of disease will shift and fly.
With extreme weather events becoming more frequent,
The consequences of global warming will be severe.

Oh, the consequences of global warming,
A menace that threatens all life and all things.
Oh, the consequences of global warming,
A menace that threatens all life and all things.
Oh, the consequences of global warming,
A menace that threatens all life and all things.
Oh, the consequences of global warming,
A menace that threatens all life and all things.
Oh, the consequences of global warming,
A menace that threatens all life and all things.
Oh, the consequences of global warming,
A menace that threatens all life and all things.
Oh, the consequences of global warming,
A menace that threatens all life and all things.
Oh, the consequences of global warming,
A menace that threatens all life and all things.
```

**Instruction 2:** *Name two items that are nearly impossible to find in stores during the pandemic.*

**CoLA:** Amidst the pandemic, when we are advised to stay indoors,
And the world outside seems so chaotic and bitter,
It's hard to find some basic necessities,
And the problem gets worse as time passes.

Two items that are hard to find,
Are the ones that you need every day,
Like toilet paper and hand sanitizer,
The shelves are empty, there's nothing to say.

These items, we used to buy without thinking twice,
Are now a luxury, hard to come by,
So let's be kind to each other and our community,
And stay safe, healthy and positive.

**Baseline:** Oh, the pandemic hath brought forth a great upheaval,
In the world of commerce, a sudden and abrupt cessation.
With the shutters down, the shelves remain empty,
Leaving us with a void that is hard to ignore entirely.

In this time of crisis, there are certain items that are scarce,
And hard to find in the markets, leaving us with a sense of despair.
One such item is the medical supplies that are in high demand,
And the other is the food items that are essential for all and all.

The medical supplies like masks, gloves, and respirators,
Are essential for the healthcare workers who are fighting the pandemicisors.
And the food items like rice, wheat, sugar, oil, and pulses,
Are essential for all Vš Vš Vš Vš Vš Vš Vš Vš Vš Vš Vš Vš
Vš Vš Vš Vš Vš Vš Vš Vš Vš Vš Vš Vš Vš Vš Vš Vš Vš Vš@nate
Vš@nate@nate@nate@nate@nate@nate@nate@nate@nate
@nate@nate@nate@nate@nate@nate@nate...

---

• **Generation on Agentic Tasks on Alfworld after RL.**

**Task** *Generation on Agentic Tasks on Alfworld after RL.*

**CoLA:** **user:**
Interact with a household to solve a task. Imagine you are an intelligent agent in a household environment and your target is to perform actions to complete the task goal. At the beginning of your interactions, you will be given the detailed description of the current environment and your goal to accomplish.

For each of your turn, you will be given the observation of the last turn. You should first think about the current condition and plan for your future actions, and then output your action in this turn. Your output must strictly follow this format:

```
Thought:  your thoughts.
Action:  your next action.

The available actions are:
1.  go to {recep}
2.  task {obj} from {recep}
3.  put {obj} in/on {recep}
4.  open {recep}
5.  close {recep}
6.  toggle {obj} {recep}
7.  clean {obj} with {recep}
8.  heat {obj} with {recep}
9.  cool {obj} with {recep}
where {obj} and {recep} correspond to objects and receptacles.
After your each turn, the environment will give you immediate feedback based
on which you plan your next few steps.  if the environment output "Nothing
happened", that means the previous action is invalid and you should try more
options.
Your response should use the following format:
Thought:  <your thoughts>
Action:  <your next action>
```
**assistant:**
OK
**user:**
You are in the middle of a room.  Looking quickly around you, you see a
bathtubbasin 1, a countertop 1, a drawer 8, a drawer 7, a drawer 6, a
drawer 5, a drawer 4, a drawer 3, a drawer 2, a drawer 1, a garbagecan 1,
a handtowelholder 1, a sinkbasin 1, a toilet 1, a toiletpaperhanger 1, a
towelholder 2, and a towelholder 1.
Your task is to:  find two tissuebox and put them in toilet.
**assistant:**
Thought:  I need to locate two tissue boxes as per the task.  The first
place to check for a tissue box is drawer 1, as it is a common storage place
for such items.
Action:  go to drawer 1
**user:**
Observation:  The drawer 1 is closed.
**assistant:**
Thought:  The drawer is closed.  I must open it to see if there's a tissue
box inside.
Action:  open drawer 1
**user:**
Observation:  You open the drawer 1.  The drawer 1 is open.  In it, you see
nothing.
**assistant:**
Thought:  Drawer 1 is empty.  I should close it and check the next drawer
for a tissue box.
Action:  close drawer 1
**user:**
Observation:  You close the drawer 1.
**assistant:**
Thought:  The next logical step is to check drawer 2 for a tissue box.

```
Action:  go to drawer 2
```
**user:**
```
Observation:  The drawer 2 is closed.
```
**assistant:**
```
Thought:  This drawer is also closed.  I'll open it to search for a tissue
box.
Action:  open drawer 2
```
**user:**
```
Observation:  You open the drawer 2.  The drawer 2 is open.  In it, you see
nothing.
```
**assistant:**
```
Thought:  No tissue box in drawer 2 either.  I should close this drawer and
continue the search.
Action:  close drawer 2
```
**user:**
```
Observation:  You close the drawer 2.
```
**assistant:**
```
Thought:  Moving on to drawer 3 to find a tissue box.
Action:  go to drawer 3
```
**user:**
```
Observation:  The drawer 3 is closed.
```
**assistant:**
```
Thought:  Another closed drawer.  I'll open it to look for a tissue box.
Action:  open drawer 3
```
**user:**
```
Observation:  You open the drawer 3.  The drawer 3 is open.  In it, you see
a candle 2.
```
**assistant:**
```
Thought:  This drawer contains a candle, not a tissue box.  I should close
it and keep looking.
Action:  close drawer 3
```
**user:**
```
Observation:  You close the drawer 3.
```
**assistant:**
```
Thought:  I'll check drawer 4 next for a tissue box.
Action:  go to drawer 4
```
**user:**
```
Observation:  On the drawer 4, you see a toiletpaper 2.
```
**assistant:**
```
Thought:  There's toilet paper on this drawer, but no tissue box.  I'll move
on to the next drawer.
Action:  go to drawer 5
```
**user:**
```
Observation:  The drawer 5 is closed.
```
**assistant:**
```
Thought:  Another closed drawer.  I'll open it to search for a tissue box.
Action:  open drawer 5
```
**user:**
```
Observation:  You open the drawer 5.  The drawer 5 is open.  In it, you see
a tissuebox 4.
```
**assistant:**
```
Thought:  I found a tissue box in drawer 5.  I need to take it to complete
the task.
```

```
Action:  take tissuebox 4 from drawer 5
```
**user:**
```
Observation:  You pick up the tissuebox 4 from the drawer 5.
```
**assistant:**
```
Thought:  With one tissue box in hand, I should close the drawer to keep the
room tidy.
Action:  close drawer 5
```
**user:**
```
Observation:  You close the drawer 5.
```
**assistant:**
```
Thought:  I should place the tissue box in the toilet as instructed.  I'll
go to the toilet to do this.
Action:  go to toilet 1
```
**user:**
```
Observation:  On the toilet 1, you see a candle 1, and a toiletpaper 1.
```
**assistant:**
```
Thought:  I'm at the toilet.  I'll place the tissue box here as required.
Action:  put tissuebox 4 in/on toilet 1
```
**user:**
```
Observation:  You put the tissuebox 4 in/on the toilet 1.
```
**assistant:**
```
Thought:  I need to find one more tissue box.  Drawer 6 is the next logical
place to check.
Action:  go to drawer 6
```
**user:**
```
Observation:  The drawer 6 is closed.
```
**assistant:**
```
Thought:  I'll open drawer 6 to see if there's another tissue box inside.
Action:  open drawer 6
```
**user:**
```
Observation:  You open the drawer 6.  The drawer 6 is open.  In it, you see
nothing.
```
**assistant:**
```
Thought:  Drawer 6 is empty.  I'll close it and continue the search.
Action:  close drawer 6
```
**user:**
```
Observation:  You close the drawer 6.
```
**assistant:**
```
Thought:  Moving on to drawer 7 to find the second tissue box.
Action:  go to drawer 7
```
**user:**
```
Observation:  The drawer 7 is closed.
```
**assistant:**
```
Thought:  I need to open drawer 7 to check for a tissue box.
Action:  open drawer 7
```
**user:**
```
Observation:  You open the drawer 7.  The drawer 7 is open.  In it, you see
nothing.
```
**assistant:**
```
Thought:  No tissue box in drawer 7.  I'll close it and check the last
drawer.
Action:  close drawer 7
```
**user:**

```
Observation:  You close the drawer 7.
```
**assistant:**
```
Thought:  The last drawer to check is drawer 8.  Hopefully, the second
tissue box is there.
Action:  go to drawer 8
```
**user:**
```
Observation:  The drawer 8 is closed.
```
**assistant:**
```
Thought:  I'll open drawer 8 to search for the second tissue box.
Action:  open drawer 8
```
**user:**
```
Observation:  You open the drawer 8.  The drawer 8 is open.  In it, you see
nothing.
```
**assistant:**
```
Thought:  Drawer 8 is also empty.  I'll close it and need to look elsewhere.
Action:  close drawer 8
```
**user:**
```
Observation:  You close the drawer 8.
```
**assistant:**
```
Thought:  I haven't checked the countertop yet.  It's a common place for
such items.
Action:  go to countertop 1
```
**user:**
```
Observation:  On the countertop 1, you see a soapbottle 1, a spraybottle 2,
a spraybottle 1, a tissuebox 3, a tissuebox 2, and a tissuebox 1.
```
**assistant:**
```
Thought:  There are tissue boxes on the countertop.  I'll take one of them
to complete the task.
Action:  take tissuebox 3 from countertop 1
```
**user:**
```
Observation:  You pick up the tissuebox 3 from the countertop 1.
```
**assistant:**
```
Thought:  With the second tissue box in hand, I should close the drawer to
keep the room tidy.
Action:  close drawer 8
```
**user:**
```
Observation:  Nothing happens.
```
**assistant:**
```
Thought:  I have both tissue boxes now.  I need to place the second one in
the toilet as instructed.
Action:  go to toilet 1
```
**user:**
```
Observation:  On the toilet 1, you see a candle 1, a tissuebox 4, and a
toiletpaper 1.
```
**assistant:**
```
Thought:  I'm at the toilet with the second tissue box.  I'll place it as
required.
Action:  put tissuebox 3 in/on toilet 1
```

**Baseline: user:**
```
Interact with a household to solve a task.  Imagine you are an intelligent
agent in a household environment and your target is to perform actions to
```

complete the task goal.  At the beginning of your interactions, you will be
given the detailed description of the current environment and your goal to
accomplish.

For each of your turn, you will be given the observation of the last
turn.  You should first think about the current condition and plan for your
future actions, and then output your action in this turn.  Your output must
strictly follow this format:

Thought:  your thoughts.
Action:  your next action.

The available actions are:
1.  go to {recep}
2.  task {obj} from {recep}
3.  put {obj} in/on {recep}
4.  open {recep}
5.  close {recep}
6.  toggle {obj} {recep}
7.  clean {obj} with {recep}
8.  heat {obj} with {recep}
9.  cool {obj} with {recep}
where {obj} and {recep} correspond to objects and receptacles.
After your each turn, the environment will give you immediate feedback based
on which you plan your next few steps.  if the environment output "Nothing
happened", that means the previous action is invalid and you should try more
options.
Your response should use the following format:
Thought:  <your thoughts>
Action:  <your next action>
**assistant:**
OK
**user:**
You are in the middle of a room.  Looking quickly around you, you see a
bathtubbasin 1, a countertop 1, a drawer 8, a drawer 7, a drawer 6, a
drawer 5, a drawer 4, a drawer 3, a drawer 2, a drawer 1, a garbagecan 1,
a handtowelholder 1, a sinkbasin 1, a toilet 1, a toiletpaperhanger 1, a
towelholder 2, and a towelholder 1.
Your task is to:  find two tissuebox and put them in toilet.
**assistant:**
Thought:  I need to locate two tissue boxes to complete the task.  I should
start by checking the nearest storage areas, which are the drawers.
Action:  go to drawer 1
**user:**
Observation:  The drawer 1 is closed.
**assistant:**
Thought:  Drawer 1 is closed.  I need to open it to see if there are any
tissue boxes inside.
Action:  open drawer 1
**user:**
Observation:  You open the drawer 1.  The drawer 1 is open.  In it, you see
nothing.
**assistant:**

```
Thought:  The drawer is empty.  I should close it and check the next drawer
for the tissue boxes.
Action:  close drawer 1
```
**user:**
```
Observation:  You close the drawer 1.
```
**assistant:**
```
Thought:  I'll move on to the next drawer to continue my search for the
tissue boxes.
Action:  go to drawer 2
```
**user:**
```
Observation:  The drawer 2 is closed.
```
**assistant:**
```
Thought:  Drawer 2 is also closed.  I need to open it to check for the
tissue boxes.
Action:  open drawer 2
```
**user:**
```
Observation:  You open the drawer 2.  The drawer 2 is open.  In it, you see
nothing.
```
**assistant:**
```
Thought:  This drawer is empty as well.  I'll close it and continue
searching the remaining drawers.
Action:  close drawer 2
```
**user:**
```
Observation:  You close the drawer 2.
```
**assistant:**
```
Thought:  I'll proceed to the next drawer to continue my search for the
tissue boxes.
Action:  go to drawer 3
```
**user:**
```
Observation:  The drawer 3 is closed.
```
**assistant:**
```
Thought:  Drawer 3 is closed.  I need to open it to look for the tissue
boxes.
Action:  open drawer 3
```
**user:**
```
Observation:  You open the drawer 3.  The drawer 3 is open.  In it, you see
a candle 2.
```
**assistant:**
```
Thought:  There's only a candle in this drawer.  I'll close it and move on
to the next drawer.
Action:  close drawer 3
```
**user:**
```
Observation:  You close the drawer 3.
```
**assistant:**
```
Thought:  I'll check the next drawer for the tissue boxes.
Action:  go to drawer 4
```
**user:**
```
Observation:  On the drawer 4, you see a toiletpaper 2.
```
**assistant:**
```
Thought:  There's a roll of toilet paper on drawer 4, but no tissue boxes.
I'll close this drawer and continue searching.
Action:  close drawer 4
```
**user:**

```
Observation:  Nothing happens.
```
**assistant:**
```
Thought:  I'll check the next drawer for the tissue boxes.
Action:  go to drawer 5
```
**user:**
```
Observation:  The drawer 5 is closed.
```
**assistant:**
```
Thought:  Drawer 5 is closed.  I need to open it to see if there are any
tissue boxes inside.
Action:  open drawer 5
```
**user:**
```
Observation:  You open the drawer 5.  The drawer 5 is open.  In it, you see
a tissuebox 4.
```
**assistant:**
```
Thought:  I found a tissue box in drawer 5.  I'll take it so I can place it
in the toilet as instructed.
Action:  take tissuebox 4 from drawer 5
```
**user:**
```
Observation:  You pick up the tissuebox 4 from the drawer 5.
```
**assistant:**
```
Thought:  With one tissue box in hand, I need to place it in the toilet as
per the task.
Action:  go to toilet 1
```
**user:**
```
Observation:  On the toilet 1, you see a candle 1, and a toiletpaper 1.
```
**assistant:**
```
Thought:  I'm at the toilet.  I should place the tissue box here to complete
part of the task.
Action:  put tissuebox 4 in/on toilet 1
```
**user:**
```
Observation:  You put the tissuebox 4 in/on the toilet 1.
```
**assistant:**
```
Thought:  I need to find the second tissue box.  I'll check the next drawer.
Action:  go to drawer 6
```
**user:**
```
Observation:  The drawer 6 is closed.
```
**assistant:**
```
Thought:  Drawer 6 is closed.  I need to open it to look for the second
tissue box.
Action:  open drawer 6
```
**user:**
```
Observation:  You open the drawer 6.  The drawer 6 is open.  In it, you see
nothing.
```
**assistant:**
```
Thought:  This drawer is empty.  I'll close it and continue searching the
remaining drawers.
Action:  close drawer 6
```
**user:**
```
Observation:  You close the drawer 6.
```
**assistant:**
```
Thought:  I'll check the next drawer for the second tissue box.
Action:  go to drawer 7
```
**user:**

```
Observation:  The drawer 7 is closed.
```
**assistant:**
```
Thought:  Drawer 7 is closed.  I need to open it to check for the second
tissue box.
Action:  open drawer 7
```
**user:**
```
Observation:  You open the drawer 7.  The drawer 7 is open.  In it, you see
nothing.
```
**assistant:**
```
Thought:  This drawer is also empty.  I'll close it and continue searching
the last drawer.
Action:  close drawer 7
```
**user:**
```
Observation:  You close the drawer 7.
```
**assistant:**
```
Thought:  I'll check the last drawer for the second tissue box.
Action:  go to drawer 8
```
**user:**
```
Observation:  The drawer 8 is closed.
```
**assistant:**
```
Thought:  Drawer 8 is closed.  I need to open it to see if there's the
second tissue box inside.
Action:  open drawer 8
```
**user:**
```
Observation:  You open the drawer 8.  The drawer 8 is open.  In it, you see
nothing.
```
**assistant:**
```
Thought:  This drawer is empty as well.  I'll close it and recheck the
previous drawers in case I missed something.
Action:  close drawer 8
```
**user:**
```
Observation:  You close the drawer 8.
```
**assistant:**
```
Thought:  I'll go back to drawer 1 to double-check for the second tissue
box.
Action:  go to drawer 1
```
**user:**
```
Observation:  The drawer 1 is closed.
```
**assistant:**
```
Thought:  Drawer 1 is closed again.  I need to open it to check for the
second tissue box.
Action:  open drawer 1
```
**user:**
```
Observation:  You open the drawer 1.  The drawer 1 is open.  In it, you see
nothing.
```
**assistant:**
```
Thought:  This drawer is still empty.  I'll close it and continue searching
the other drawers.
Action:  close drawer 1
```

• **Generation on Math Reasoning with MCTS-Q.** We show the generations between CoLA using MCTS-Q, baseline

using MCTS-Q and CoLA (using FTA-P).

---

**Problem 1:** *In how many ways can 8 people sit around a round table if 3 of the people -- Pierre, Rosa, and Thomas -- all want to sit together? (Two seatings are considered the same if one is a rotation of the other.)*

**Answer:** 720

**CoLA with MCTS-Q:** 1. **Group Pierre, Rosa, and Thomas:** Treat Pierre, Rosa, and Thomas as a single entity. This reduces the problem to arranging 6 entities (the group and the other 5 individuals) around the table.

2. **Arrange the 6 entities:** There are 6! ways to arrange these 6 entities around the table.

3. **Arrange the individuals within the group:** There are 3! ways to arrange Pierre, Rosa, and Thomas within their group.

4. **Account for rotations:** Since the table is round, each arrangement can be rotated in 6 different ways (one for each person as the starting point). Therefore, we need to divide by 6 to correct for these rotations.

5. **Calculate the total number of arrangements:**
$\frac{6! \times 3!}{6} = \frac{720 \times 6}{6} = 720$

The final answer is 720 .

**CoLA (SFT):** 1. **Calculate the total number of ways to arrange 8 people around a round table:**
Since the table is round, we can fix one person's position to avoid counting rotations as different arrangements. This gives us 7! ways to arrange the remaining 7 people.

2. **Calculate the number of ways to arrange Pierre, Rosa, and Thomas together:**
Treat Pierre, Rosa, and Thomas as a single unit. This reduces the problem to arranging 6 units (the unit of Pierre, Rosa, and Thomas, and the other 5 individuals) around the table. Fixing one of these units, we have 5! ways to arrange the remaining 5 units.

3. **Calculate the number of ways to arrange Pierre, Rosa, and Thomas within their unit:**
Pierre, Rosa, and Thomas can be arranged among themselves in 3! ways.

4. **Combine the arrangements:**
Multiply the number of ways to arrange the 6 units by the number of ways to arrange Pierre, Rosa, and Thomas within their unit:
$5! \times 3! = 120 \times 6 = 720$

5. **Adjust for the round table:**
Since the table is round, we need to divide by the number of people (8) to avoid overcounting rotations:
$\frac{720}{8} = 90$

The final answer is 90 .

---

**Baseline with MCTS-Q** We'll consider the three people who want to sit together as a single unit. That leaves us with 5 units to arrange around the table (the 3-person unit and the other 5 individuals).

1. Calculate the number of ways to arrange these 5 units around the table. Since they are indistinguishable in terms of rotation, we use the formula for circular permutations of $n$ items, which is $(n-1)!$. Here, $n=5$:
Ways to arrange 5 units $=(5-1)!=4!$
$4!=4\times 3\times 2\times 1=24$

2. Now, within the 3-person unit, Pierre, Rosa, and Thomas can be arranged among themselves in $3!$ ways:
$3!=3\times 2\times 1=6$

3. Multiply the number of ways to arrange the units by the number of ways to arrange the individuals within the unit to get the total number of arrangements:
Total arrangements $=24\times 6=144$

$\boxed{144}$

**Problem 2:** *If* $0.\overline{1331}$ *is written as a fraction* $frac{a}{b}$ *with* $a$ *and* $b$ *being positive integers with a greatest common divisor of 1, what is* $a+b$?

**Answer:** $\boxed{1030}$

**CoLA with MCTS-Q:** 1. Let $x = 0.\overline{1331}$. Then, multiplying both sides by 10000 (since the repeating part has four digits), we get:
$10000x = 1331.\overline{1331}$
2. Subtract the original $x$ from this new equation:
$10000x - x = 1331.\overline{1331} - 0.\overline{1331}$
$9999x = 1331$
3. Solving for $x$, we find:
$x = \frac{1331}{9999}$
4. To simplify $\frac{1331}{9999}$, we find the greatest common divisor of 1331 and 9999. Since 9999 is $3^2 \times 11 \times 101$ and 1331 is $11^3$, the gcd is 11. Thus, we can simplify the fraction:
$\frac{1331}{9999} = \frac{1331 \div 11}{9999 \div 11} = \frac{121}{909}$
5. The sum of the numerator and the denominator is:
$121 + 909 = \boxed{1030}$

**CoLA (SFT):** 1. Let $x = 0.\overline{1331}$. To eliminate the repeating decimal, multiply $x$ by 10000 (since the repeating part has four digits):
$10000x = 1331.\overline{1331}$
2. Subtract the original $x$ from this equation:
$10000x - x = 1331.\overline{1331} - 0.\overline{1331}$
$9999x = 1331$
3. Solve for $x$:
$x = \frac{1331}{9999}$
4. Simplify the fraction $\frac{1331}{9999}$ by finding the greatest common divisor (GCD) of 1331 and 9999. Since 1331 is a perfect square $11^3$ and 9999 is not a multiple of 11, the GCD is 1. Thus, the fraction is already in its simplest form.
5. Calculate $a+b$ where $a = 1331$ and $b = 9999$:
$a + b = 1331 + 9999 = \boxed{11330}$

**Baseline with MCTS-Q** Let $x = 0.\overline{1331}$. Then $1000x = 1331.\overline{1331}$, so $1000x - x = 1331.\overline{1331} - 0.\overline{1331}$, which simplifies to $1000x - x = 1331$. Solving for $x$, we get $999x = 1331$, so $x = \frac{1331}{999}$.

To simplify $\frac{1331}{999}$, we find the greatest common divisor of the numerator and the denominator. Since $999 = 3^3 \times 37$ and $1331 = 11^3$, the greatest common divisor of 999 and 1331 is 1. Thus, the fraction is already in its simplest form.

Therefore, $a + b = 1331 + 999 = \boxed{2330}$.

