# OpenReview forum: "Controlling Large Language Model with Latent Action"
_ICML.cc/2025/Conference — ICML 2025 poster_

### Official Review · Reviewer_mZHY · 2025-03-04

**Overall Recommendation:** 3

**Summary:**

Existing LLMs often rely on token-level actions that may be overly large and inefficient. This paper proposes learning a compact and latent action space to improve controllability and exploration in RL. Specifically, the authors augment a latent action space with a pre-trained LLM to form a latent action language world model. This latent action model is extensively trained from the pre-training dataset and tuned in the post-training stage. Experiments with Llama-3.1-8B as the base model demonstrate that using this latent action model for RL training achieves better controllability on multiple tasks, including multiple preference alignment tasks, math reasoning task, as well as the improved and flexible searching over the token-level action framework.

**Claims And Evidence:**

Most claims are supported by clear evidence.

Some are not:
1) Therefore, LAMP aims to decouple these semantics by assigning them to different and limited latent actions, enabling more efffcient ffne-tuning, alignment, and search by latent action control.   ==> I did not see such evidence.

2) However, in our experiments, we observed that this suffers from codebook collapse, where only a limited number of actions are activated during training.

**Essential References Not Discussed:**

NA

**Experimental Designs Or Analyses:**

Yes. The four questions investigated are important and the experimental designs are reasonable.

However, the authors should show that their methods are appliable to other LLMs besides Llama-3.1-8B.

**Methods And Evaluation Criteria:**

Yes.

**Other Comments Or Suggestions:**

‘’RL from observations only‘’ ==> ``RL from observations only”
In this section, we verify our framework across multiple training stages. First, we introduce the basic setting of our model design. Then we validate LAMP in multiple training stages.  ==> is redundant
In the generation process, we compute the next token from  the world model in a greedy manner. We only randomly or greedily sample actions from the policy model.  ==> what's your meaning?

**Other Strengths And Weaknesses:**

Strengths:
The latent action is a very interesting topic, and I think this is important for the future of LLMs
The methods are reasonable, and effective.
The paper is generally well-written, easy-understanding.

Weaknesses:
small writing issues, see [Other Comments Or Suggestions]
I did not see the necessary of MCTS-Q? It makes the methods complex?
The results did not show that the proposed LAMP are better than baselines with a clear margin.

**Questions For Authors:**

line-199: why the current time embedding e^i_t has a superscript i?
I did not see the necessary of MCTS-Q? It makes the methods complex?
The results did not show that the proposed LAMP are better than baselines with a clear margin.

**Relation To Broader Scientific Literature:**

The effectiveness of latent action will be related to most of the LLM researchers. Especially those have RL background.

**Theoretical Claims:**

NA

---

> ### Author Rebuttal · Authors · 2025-04-01
>
> We sincerely appreciate the reviewer's valuable feedback and recognition of our latent action approach.
>
> **Q1:** Claims are not supported by clear evidence: Therefore, LAMP aims to decouple these semantics by assigning them to different and limited latent actions, enabling more efficient fine-tuning, alignment, and search by latent action control. ==> I did not see such evidence.
>
> **A1:** For claim 1, we realize that this claim may potentially lead to misunderstandings and would like to clarify. Figure 12 and Figure 10 (Appendix C.2) aim to show that latent actions have higher semantics than tokens. We will revise the statement in Section 4 to clarify that the connection between latent actions and semantics is weakly correlated, which should avoid misunderstandings.
>
> **Q2:** Claims are not supported by clear evidence: However, in our experiments, we observed that this suffers from codebook collapse, where only a limited number of actions are activated during training.
>
> **A2:** For claim 2, we compare VQ-VAE with our proposed method in Figure 8 (Appendix C.1) by counting the number of actions with non-zero usage. Results show that VQ-VAE suffers from severe codebook collapse, while our approach effectively maintains the utilization of codes in the codebook. We will add a reference to this claim to make it more supported.
>
> **Q3:** However, the authors should show that their methods are appliable to other LLMs besides Llama-3.1-8B.
>
> **A3:**  We have implemented the LAMP architecture with Qwen-2.5-math-1.5B (Appendix C.1, Figure 9). We observed similar behavioral patterns to those in Llama-3.1-8B, including stable loss reduction and maintained codebook diversity without collapse. This provides preliminary evidence for LAMP's generalizability. However, due to the limitation of computation resources, we did not proceed to full-scale training and subsequent downstream task optimization.
>
> **Q4:** In the generation process, we compute the next token from the world model in a greedy manner. We only randomly or greedily sample actions from the policy model. ==> what's your meaning?
>
> **A4:** This sentence describes LAMP's language generation process. During the RL training phase, LAMP's policy model explores by randomly sampling latent actions, while during the evaluation phase, it deterministically (greedily) samples latent actions to ensure consistent results by selecting the maximum probability action. We will further revise here to make a clearer description.
>
> **Q5:** Some writing issues, including typos and redundant descriptions.
>
> **A5:** We will improve the writing in the revision.
>
> **Q6:** line-199: why the current time embedding e^i_t has a superscript i?
>
> **A6:** Here we aim to differentiate the embedding from other model input/output embeddings, and $i$ represents the inverse dynamics model.
>
> **Q7:** I did not see the necessary of MCTS-Q? It makes the methods complex?
>
> **A7:** MCTS-Q demonstrates how our latent action formulation can effectively integrate techniques that were previously difficult to apply in token space. For instance, Q-learning-based uncertainty estimation is a key technique for pruning the search space in MCTS, but scaling it in token space is challenging. Thanks to our latent action space design, however, this approach now shows promising results. Importantly, MCTS-Q does not introduce significant complexity—instead, it makes MCTS more effective.
>
> **Q8:** The results did not show that the proposed LAMP are better than baselines with a clear margin.
>
> **A8:** We guess the reviewer draws this conclusion from the results in Table 1 and Table 2. We want to clarify the performance gains of our approach. Our method primarily demonstrates superior downstream performance in multiple tasks. **For preference alignment**, we achieve an average win rate of **64%** across four tasks and exceed **90%** under the KL=0.00 setting. **For math reasoning**, we also shows significant improvement (**42.6** vs 38.2). **For search**, combined MCTS with Q, our method can achieve better performance (**68.2** vs 63.0). **We also explore an agentic task called Alfworld**[1], which is an embodied multi-turn interactive task. With the same fine-tuning and RL scheme, results shown in following table demonstrate superior performance of LAMP on it:
>
> | Method | Seen | Unseen |
> |-|-|-|
> |Llama-3.1-8B-SFT|68.6|67.9|
> |Llama-3.1-8B-RL|68.6|71.6|
> |LAMP-FTA|75.7|70.9|
> |LAMP-RL|**77.9**|**74.6**|
> We also need to clarify that Tables 1 and 2 show only "me too", which only shows the comparable initialized performance of LAMP. We will revise the descriptions in Section 5.1 and Section 5.2 to strengthen the presentation of these aspects. Please let us know if you have additional concerns about the performance gains of our approach.
> [1] Shridhar et al. ALFWorld: Aligning Text and Embodied Environments for Interactive Learning. arXiv:2010.03768 (2020).

---

### Official Review · Reviewer_X62N · 2025-03-14

**Overall Recommendation:** 3

**Summary:**

The paper presents Latent Action governed world Model from Pre-trained LLM (LAMP), a novel reinforcement learning (RL) approach designed to control Large Language Models (LLMs) by learning a compact, latent action space rather than relying on conventional token-level actions. LAMP leverages an inverse dynamics model to create discrete latent actions and a policy model to guide these actions for downstream tasks. Experimental results on the Llama-3.1-8B model show moderate improvements in performance.

**Claims And Evidence:**

While the claims regarding improved controllability is generally supported, the main concern is that the performance gains demonstrated by LAMP are relatively marginal compared to baseline methods. Specifically, the reported improvements such as Table 1 and Table 2, appear limited in practical significance given the complexity of the proposed method.

Additionally, the claim about the semantic interpretability of latent actions is not fully convincing. It remains unclear how these high-level semantic categories meaningfully translate into effective control or actionable improvements.

**Essential References Not Discussed:**

This paper appears comprehensive in its coverage of related works and does not omit essential references critical to understanding the contributions.

**Experimental Designs Or Analyses:**

The experimental design and analyses appear generally sound.

**Methods And Evaluation Criteria:**

- The proposed methods make sense and the motivation is clear.
- The connection between LLM training and observation-only RL is novel.
- The evaluation criteria are reasonable.

**Other Comments Or Suggestions:**

N/A

**Other Strengths And Weaknesses:**

N/A

**Questions For Authors:**

N/A

**Relation To Broader Scientific Literature:**

The paper situates itself effectively within the broader scientific literature, clearly outlining the limitations of token-level control and the motivation drawn from "RL from observations" literature. The contributions are well-contextualized relative to recent developments such as RLHF and latent-space modeling.

**Theoretical Claims:**

The paper does not present theoretical claims.

---

> ### Author Rebuttal · Authors · 2025-04-01
>
> We sincerely appreciate the reviewer's thoughtful feedback and positive recognition of the novelty of the perspective of RL from observation. We are grateful for the opportunity to address the reviewer's concerns and further clarify our methodology.
>
> **Q1:** While the claims regarding improved controllability is generally supported, the main concern is that the performance gains demonstrated by LAMP are relatively marginal compared to baseline methods. Specifically, the reported improvements such as Table 1 and Table 2, appear limited in practical significance given the complexity of the proposed method.
>
> **A1:** We apologize for the unclear expression. We clarify that **Tables 1 and 2 show only "me too" but not the main purpose of LAMP**, which is to enhance the downstream tasks using reinforcement learning.
>
> > Explanation: Table 1 shows the performance of LAMP model with behavior cloning policy, which should be called as LAMP-BC. We train LAMP-BC just to show that the LAMP structure can perform as an ordinary LLM, as presented results in Table 1. Further, Table 2 shows supervised fine-tuned performance of both the base LLM and LAMP-BC, which shows LAMP-BC can have a slightly better performance. We will revise the corresponding descriptions in Sections 5.1 and 5.2 to strengthen the presentation for the propose of these two experiments to explain this more clearly.
>
> To provide an overview of the power of our approach, we summarize and explain our RL results below:
>
> - For preference alignment tasks, our method achieves superior win rates and robustness via RLHF across different KL settings. In **Figure 2**, our LAMP model achieves an average of 64% win rate at KL=0.01 and over 90% win rate at KL=0.00. In **Figure 3**, our LAMP model achieves a slight performance improvement under the KL=0.00 setting compared to the standard KL=0.01 configuration, indicating a more robust alignment of our LAMP model.
> - For mathematical reasoning, after fine-tuning and RL optimization, our method achieves 42.6 on MATH500, comparing with 38.2 of baseline. And achieves 68.2 on a more flexible MCTS method, comparing with 63.0 of baseline.
>
> These results show that **our method achieves significant improvements by introducing latent actions for RL training**.
>
> **Q2:** Additionally, the claim about the semantic interpretability of latent actions is not fully convincing. It remains unclear how these high-level semantic categories meaningfully translate into effective control or actionable improvements.
>
> **A2:** We would like to clarify that **we did not claim the semantic interpretability of the extracted latent actions**. In fact, Figure 12 (Appendix C.2, Page 18) aims to show that latent actions have higher semantics than tokens. We will revise the claim in Section 4 to state that the connection of the latent actions to semantic is weakly correlated, avoiding cause misunderstandings.
>
> To address your concern about the effectiveness of latent actions, we clarify that the higher-level latent actions not only enhance the reinforcement learning performance due to reduced latent action space, but also mitigate the reward hacking issue (Figure 3), which can lead to more stable reinforcement learning.

---

### Official Review · Reviewer_iaGP · 2025-03-16

**Overall Recommendation:** 3

**Summary:**

This paper proposes to learn a more compact latent "action" space for pretrained LLMs to improve controllability. This latent action space is learned by an autoencoder where the encoder is an inverse dynamics model p(a_t|x_0:t+1), the decoder is a transition model p(x_t+1|x_0:t, a_t), the conditional prior is a policy p(a_t|x_0:t). The training objective for this latent action space is similar to VQ-VAE, but parametrize p(a_t|x_0:t+1) with Gumbel Softmax and the KL term in the ELBO is parametrized by the entropy of p(a_t|x_0:t+1). The authors further provide two SFT approaches based on finetuning either the inverse dynamics or the policy, as well as a RL objective on the latent policy. Experiments show that the proposed method can induce a latent space with only a slight degradation of the base pretrained model. The authors further show that RLHF and SFT over latent actions appear to be more effective than on the base model with tokens as actions.

### After rebuttal ###
Thanks the authors for addressing my concerns. I will keep my initial accepting rating.

**Claims And Evidence:**

The thesis claim is that the latent action, due to its lower dimensionality, can be a more controllable alternatives to the token space. The authors provide evidence that SFT and RLHF on the latent space turn out to perform better.

**Essential References Not Discussed:**

N/A

**Experimental Designs Or Analyses:**

The authors first validate the success of latent space induction by showing that there is only a slight degrade of the performance after learning the latent space modules.
They then test how well the latent actions control the tokens generation.
The main results are the performance gain from SFT and RLHF on the latent action space.
The authors also provided ablation study for the increased token and parameter numbers. However, this appears to be skeptical to me. The reported ablation show degradation of the base model in terms of increased model size and data size, which is very counter-intuitive.

**Methods And Evaluation Criteria:**

The proposed method follows the standard framework of VQ_VAE. The adopted evaluation criteria is commonly used.

**Other Comments Or Suggestions:**

N/A

**Other Strengths And Weaknesses:**

The idea of learning latent action space to improve controllability of LLM appears to be novel to me.

**Questions For Authors:**

In the ablation study, why did increasing the model size and the data size lead to worse performance? This doesn't seem to obey the scaling laws of LLMs. I hope the authors would like to provide some explanation.

**Relation To Broader Scientific Literature:**

The idea of using a more compact space for SFT and RLHF can improve the efficiency of the SOTA pipeline.

**Theoretical Claims:**

There is no theoretical claim.

---

> ### Author Rebuttal · Authors · 2025-04-01
>
> We sincerely thank the reviewer for helpful feedback and positive assessment of the novelty in our latent action control approach. We address the reviewer's concern below, with key claims highlighted in bold for clarity.
>
> **Q1:** In the ablation study, why did increasing the model size and the data size lead to worse performance? This doesn't seem to obey the scaling laws of LLMs. I hope the authors would like to provide some explanation.
>
> **A1:** Thank you for your question. We’d like to clarify that our findings **do not conflict with the scaling laws of LLMs**. The slight performance degradation observed in Figure 6 stems from the "forgetting issue" during continued pre-training of Llama-3.1-8B. This occurs because our experimental data differs from Meta’s original pre-training data, leading to partial knowledge forgetting. Similar effects have been documented in prior work [1]. To our knowledge, scaling laws [2] are primarily applied to large-scale pre-training, not continued training with limited data.
>
> We clarify that we show Figure 6 to indicate that the original Llama-3.1-8b is a proper baseline. We will revise the descriptions to modify the explanation of this experiment in Section 5.4 to explain it more clearly. We understand that the reviewer may expect improved performance with more data. While this could be possible by enhancing LAMP with additional insights and techniques from continued training, such an investigation is beyond the scope of this paper. We appreciate your feedback and consider this an important direction for future work.
>
> [1] Shao et al. DeepSeekMath: Pushing the Limits of Mathematical Reasoning in Open Language Models. arXiv:2402.03300 (2024).
>
> [2] Kaplan et al. Scaling Laws for Neural Language Models. arXiv:2001.08361 (2020).

---

### Decision · Program_Chairs · 2025-05-01

**Decision:**

Accept (poster)

**Comment:**

This paper proposes a novel approach - connecting LLMs to learning from observation - and uses latent actions to steer a language model. This is a novel idea, and may be of interest to the community, especially given the number of very similar ideas that are now being pursued in the era of LLMs. That being said, experiments remain narrow with only Llama 3.1 8B, and it remains unclear how applicable the approach is to more open ended problems.